

# Effect of winds and waves on salt intrusion in the Pearl River Estuary

Wenping Gong[1,2], Zhongyuan Lin[1,2], Yunzhen Chen[1,2], Zhaoyun Chen[1,2], Heng Zhang[1,2,3]

[1]School of Marine Science, SunYat-sen University, Guangzhou, 510275, China

[2]Guangdong Provincial Key Laboratory of Marine Resources and Coastal Engineering, Sun Yat-sen University, Guangzhou, 510275, China

[3]Guangdong Provincial Key Laboratory for Climate Change and Natural Disaster Studies, Sun Yat-sen University, Guangzhou, 510275, China

*Correspondence to*: Heng Zhang (zhangheng@mail.sysu.edu.cn)

**Abstract.** Salt intrusion in the Pearl River Estuary (PRE) is a dynamic process that is influenced by a range of factors and to date, few studies have examined the effects of winds and waves on salt intrusion in the PRE. We investigate these effects using the Coupled-Ocean-Atmosphere-Wave-Sediment Transport (COAWST) modeling system applied to the PRE. After careful validation, the model is used for a series of diagnostic simulations. It is revealed that the local wind considerably strengthens the salt intrusion by lowering the water level in the eastern part of the estuary and increasing the bottom

landward flow. The remote wind increases the water mixing on the continental shelf, elevates the water level on the shelf and in the PRE, and pumps saltier shelf water into the estuary by Ekman transport. Enhancement of the salt intrusion is comparable between the remote and local winds. Waves decrease the salt intrusion by increasing the water mixing. Sensitivity analysis shows that the axial down-estuary wind, is most efficient in driving increases in salt intrusion via wind straining effect.

## 1 Introduction

The Pearl River, in the south of China, is the third largest river in China. The Pearl River Estuary (PRE) is the largest estuary in the Pearl River Delta (PRD) (Fig. 1a). The PRD has a population of 40 million and a booming economy. Salt intrusion in the PRE has been a major environmental concern over recent years. Salt intrusion is most serious during the dry season from November to the following March, and water is contaminated with salt to such a degree that the fresh water supply is

hampered. Meanwhile, as salt intrusion affects the estuarine density gradient, circulation and mixing processes (MacCready and Geyer, 2010; Geyer and MacCready, 2014), it affects the residence time of nutrients and contaminants (Ren et al., 2014; Sun et al., 2014); flocculation, resuspension and trapping of fine sediments (Ren et al., 2014) and phytoplankton dynamics (Lu and Gan, 2015) in the estuary. Enhanced salt intrusion also threatens the survival of some aquatic species in the estuary. It is expected that salt intrusion will be exacerbated along with sea level rise induced by climate change (Yuan et al., 2015).

Salt intrusion in an estuary is subject to many external forcings. Besides river discharge, tides and topography, winds (e.g., Ralston et al., 2008; Uncles and Stephens, 2011) and waves can also affect the extent and duration of salt intrusion. The effect of winds can be separated into remote and local wind effects (Wong and Moses-Hall, 1998; Janzen and Wong, 2002; Wong and Valle-Levinson, 2002). Herein the remote wind is referred to as that acting on the continental shelf, while the local wind is that acting inside the estuary. The remote wind usually generates a uni-directional subtidal transport across the

entrance of the estuary through the impingement of coastal sea level. On the continental shelf, downwelling-favorable winds generate a water level setup, increase mixing, drive onshore transport and pump high salinity water from the shelf into the estuaries (Allen and Newberger, 1996; Hickey et al., 2002), while upwelling-favorable winds generate a water level setdown, drive offshore transport at the water surface and onshore transport at the bottom, leading to elevated bottom salinity at the estuaries' mouth (Zu and Gan, 2015).





The local wind effect is further separated into two components: wind straining and wind mixing (Scully et al., 2005). The down-estuary wind can enhance stratification and estuarine circulation via wind straining, and can also simultaneously increase vertical mixing via wind mixing; therefore its effect on salt transport reflects the competition between these two components. However, the up-estuary wind increases vertical mixing and decreases estuarine circulation, consequently
reducing the salt intrusion (Chen and Sanford, 2009; Jia and Li, 2012).

Along with winds, waves can have an important role in modifying salt intrusion. Waves can generate wave-induced circulation, and change the mixing in the water column, especially at the water surface because of wave breaking, and in the bottom boundary layer through wave-current interactions. Gerbi et al. (2013) investigated the effect of breaking surface waves on salinity changes in a river plume during upwelling-favorable winds. Rong et al. (2014) studied the effect of wave-
current interactions on the Mississippi-Atchafalaya river plume and the salinity variations on the Texas-Louisiana shelf. Delpey et al. (2014) noted that in a small bay south of the French Atlantic Coast, wave-induced circulation retains the freshwater inside the bay and significantly reduces the outflow of freshwater through the bay inlets. It is evident therefore that wind waves can modify estuarine hydrodynamics and also salt intrusion. However, to the best of our knowledge, few studies have examined the effect of waves on salt intrusion.

The effects of river discharge, tides, sea level rise, and bathymetric change on salt intrusion in the PRE have been extensively investigated in numerous studies (e.g. Gong and Shen, 2011; Yuan et al., 2015; Yuan and Zhu, 2015). As salt intrusion is most serious during the dry season when strong winds and waves occur, there is a need for an improved understanding of the effects of winds and waves. The wind effect in the PRE reflects the combination of the remote and local winds. There is a saying among the local people that "the N winds drive salt intrusion"; here the wind directions are those
where they come from, and the N winds include the N, NE, and NW winds, which are all down-estuary ones. Mo et al. (2007) used a 1-D model to investigate the wind effect on salt intrusion in the tidal river of the PRE, and their results showed that salt intrusion decreases under all the down-estuary (N, NE), across-estuary (E) and up-estuary (SE) winds. Lai et al. (2016) examined the local wind effect on the Pearl River plume dynamics during the dry season and showed that salt intrusion is greatly enhanced by the downwelling-favorable wind, especially when the tidal effect is excluded. However, mechanisms of
salt intrusion enhancement by winds, in terms of both the wind magnitude and direction, have not been fully explored. Further, the effect of waves on salt intrusion in the PRE has not been investigated.

In this study, we used the Coupled-Ocean-Atmosphere-Wave-Sediment Transport (COAWST) (Warner et al., 2010) modeling system in combination with field measurements to examine the effects of winds and waves on salt transport and salt intrusion in the PRE. The main objectives of this study are: 1) to identify the relative importance of winds and waves on
salt intrusion in the PRE; and 2) to explore the processes and mechanisms behind these effects.

This paper includes a description of the study area, followed by details of how the COAWST modeling system is implemented. We have documented the model validation process and the various simulations we used to investigate salt intrusion. In these simulations, we consider the effects of local and remote winds, wind directions, and waves. We also identify, calculate and compare the intensities, patterns and mechanisms of these effects.

**2 Study area**

The Pearl River is comprised of the West, North, and East River branches (Fig.1a). These branches converge in a large delta, the PRD, and form an extremely complex river network, the Pearl River Network, and three estuaries known, from the east to west, as the Pearl River, Modaomen and Huangmaohai estuaries. The PRE links the Pearl River Network and the continental shelf of the Northern South China Sea (NSCS). The PRE is funnel-shaped; its width decreases from 50 km at its
mouth between Hong Kong and Macau to 6 km at Humen. The axial length of the estuary is approximately 70 km (Fig.1b). The bathymetry of the estuary is complicated, and is characterized by three shoals (West Shoal, Middle Shoal and East Shoal)





and two channels (West Channel and East Channel). These three shoals are < 5 m deep, the West Channel is > 10 m deep, and the East Channel is 5–15 m deep. The tidal river extends to ~50 km beyond the estuary head (Humen Outlet, Fig.1c).

Based on Zhao (1990), the Pearl River has an annually mean river discharge of approximately 11000 $m^3/s$. The mean discharge shows distinct seasonal variations, and decreases from 20000 $m^3/s$ during the wet season from May to September

to 4000 $m^3/s$ during the dry season. Sixty-four percent of the total discharge of the Pearl River enters the PRE from four outlets, namely, from north to south, the Humen (25%), Jiaomen (13%), Hongqili (11%) and Hengmen (15%) Outlets (see Fig. 1b). The estuary has a mixed semidiurnal tidal regime, and the $M_2$ constituent dominates, followed by the $K_1$ and $O_1$ constituents (Mao et al., 2004). The mean tidal range increases from 1.45 m at the estuary mouth to 1.77 m at the Humen Outlet because of geometry convergence, and the $M_2$ tidal constitute is the most amplified. During the dry season, the PRE

is generally in a partially mixed state (Wong et al., 2003).The horizontal salinity difference varies by between 20 and 25 ppt across a distance of 70 km and the vertical salinity difference between the surface and bottom varies from 1 to 12 ppt.

The climate in the PRE is subtropical, and monsoon winds prevail. The winter is characterized by the Asian winter monsoon, and NE winds dominate (Wong et al., 2003). Analysis of 18-year meteorological data (http://www.weather.gov.hk) at the Henglan Island, Hong, shows that, in winter, the NE wind occurs most frequently (42%), followed by the E (18%), and N

(12%) winds. The NE wind has a mean speed of 5.7 m/s, while the E and N winds have mean speeds of 7.8 and 7.0 m/s, respectively. The winter monsoon is downwelling-favorable on the continental shelf, and is down-estuary inside the estuary. Wave observation data collected by the South China Sea Branch, State Oceanic Administration, China, from an offshore station outside the PRE with a water depth of 15 m indicates that swell waves predominate. During the winter, the mean significant wave height is 1.1 m, and the corresponding mean wave period is 5.5 s. The maximum wave height can be up to 4

or 5 m under strong winds. Waves from the SE dominate and occur more than 50% of the time. Waves from other directions occur less frequently; for example, ESE and SW waves each account for approximately 10% of the time.

## 3 Methdology

### 3.1 Implementation of COAWST in the PRE

We used the COAWST modeling system (version 3.2), which is an agglomeration of open-source modeling components that can be used to investigate coupled processes of atmosphere, water current, and waves in coastal ocean. In this study, we activate coupling only between ocean circulation and waves.

In the COAWST system, the ocean circulation model is the Regional Ocean Modeling System (ROMS) model (Haidvogel et al., 2000; Shchepetkin and McWilliams, 2005). We use a model domain that covers the entire PRD and part of the

continental shelf in the NSCS (Fig. 1a). An orthogonal curvilinear coordinate system is designed to follow the coastline. The grid spacing is less than 200 m in the Pearl River Network and in most of the estuaries, and decreases towards offshore. Close to open ocean boundaries, the spatial resolution is as large as 10 km. The total number of grid points is 627×546. In the vertical, 15 layers are utilized with higher resolution close to both the surface and the bottom. We use the Generic Length Scale (GLS) turbulence closure scheme to calculate the vertical mixing (Warner et al., 2005). When there are no waves, a

log-layer bottom stress is exerted at the bed. When the wave effect is included, the wave-current bottom boundary layer dynamics is activated. The horizontal eddy viscosity and diffusivity are calculated with the Smagorinsky scheme (Smagorinsky, 1963).

The open ocean boundary condition for the barotropic component consists of a Chapman/Flather boundary condition for depth-averaged flow and sea surface elevation (Chapman, 1985; Flather, 1976). The open boundary conditions for the

temperature, salinity and baroclinic current are Orlanski-type radiation condition (Orlanski, 1976). The tidal water level and





depth-averaged tidal velocity are obtained from the TPXO database (Egbert and Erofeeva, 2002), including 9 tidal constituents of $P_1$, $Q_1$, $O_1$, $K_1$, $M_2$, $S_2$, $N_2$, $K_2$ and $M_4$, while the subtidal water level and depth-averaged velocity are derived from the HYbrid Coordinate Ocean Model (HYCOM) model outputs (http://hycom.org/hycom).

Freshwater inputs from the West, North, and East Rivers are specified using daily data from the upstream hydrological

stations. Shortwave radiation, air temperature, humidity, and atmosphere pressure are specified at the water surface from the Comprehensive Ocean Atmosphere Data Set (COADS) data. Wind forcing is derived from the NCEP (National Centers for Environmental Prediction) Climate Forecast System Reanalysis (CFSR) hourly data. The wind stress is calculated using the method of Large and Pond (1981). The Coupled Ocean Atmosphere Response Experiment (COARE) 3.0 bulk flux algorithms are used to estimate the heat flux in ROMS.

The wave model in the COAWST system is the Simulating Waves Nearshore (SWAN, Booij et al., 1999). In our simulations, the SWAN and ROMS models are coupled to the same grid. Twenty-five frequencies (0.01–1 Hz) and thirty-six directional bands are used. The SWAN model boundary is obtained from the output of a large domain WAVEWATCHIII wave model that covers the entire South China Sea (SCS). This wave model is one component of the Experimental Platform of Marine Environmental Forecasting System, maintained by State Key Laboratory of Tropical Marine Environments, South China Sea

Institute of Oceanology, Chinese Academy of Science (http://210.77.90.43/). The data output from the large SCS domain is every 3 h. Local waves are generated by the wind field, which is the same as that used in the ROMS model. The SWAN model then provides the wave properties (significant wave height, mean wave direction, surface peak wave period, mean wavelength, wave energy dissipation, percent of breaking waves) to input into ROMS for estimation of the bottom shear stress, wave-induced momentum flux, and wave mixing.

As outlined in Rong et al. (2014), the effects of surface waves on circulation are manifested through three physical mechanisms, namely wave-enhanced bottom shear stress, wave-enhanced mixing, and excessive momentum flux within the circulation because of the presence of waves. To simulate the wave-enhanced bottom shear stress, we use the wave-current bottom boundary layer model of Madsen (1994) known as SSW_BBL in COAWST (Warner et al., 2008). The GLS mixing model is used to incorporate the wave-induced mixing and WDISS_THORGUZA is activated to specify the wave-induced

turbulent kinetic energy from wave breaking and bottom friction (Thorton and Guza, 1983). We activate WEC_VF and use the vortex force method proposed by McWilliams et al. (2004) and later implemented in COAWST by Kumar et al. (2012) to simulate excessive momentum flux due to waves.

Different timesteps are used for ROMS and SWAN in the simulations. In ROMS, a 5 s baroclinic time step is used with a model-splitting ratio of 10. SWAN is run in non-stationary mode with a time step of 300 s. ROMS and SWAN are coupled

synchronously with a 1-hour time interval.

To test the model's performance in the PRE, comprehensive model validation is carried out. The model validations are implemented for two periods, from September 20 to November 10, 2005, and from November 1 to December 31, 2009. The time periods are selected by the availability of observation data. We use the period from November 1 to December 31, 2009 for our diagnostic simulations, as it is more representative of the dry season.

After model validation, eight diagnostic simulations are designed and run. Case 1, includes the tides and river discharge. Case 2 includes tides, river discharge, and wind applied at the water surface inside the estuary, whose extent is shown in Fig.1b. Remote wind is activated in Case 3, in which wind forcing is specified at the surface of the continental shelf, together with specification of subtidal water level and current at the open ocean boudaries. Case 4 includes both the local and remote winds, while Case 5 includes the wave effect and the local and remote winds. In Cases 6, 7 and 8, we investigate changes in

the salt intrusion in response to different wind directions (N, NE, and E, respectively) and the associated waves. The configurations of all the model runs are listed in Table 1.



### 3.2 Definition of the length of the salt intrusion and decomposition of the salt flux

We define the length of the salt intrusion as the distance from the estuary mouth to the landward limit of the bottom 10 ppt isohaline along the longitudinal transect T1 (Fig.1c). We choose the 10 ppt isohaline because it is generally located around the estuary head, and is minimally influenced by the complex interactions that occur among river networks further upstream.

We explore the mechanisms that control wind- and wave-induced changes in salt intrusion by examining the salt transport flux at cross section T2 at the estuary mouth as this transect is the main entrance for exchange between the estuary and shelf (Fig. 1c). Another entrance lies in the eastern part of the estuary and is relatively small. The salt transport flux at T2 is calculated using the method of Lerczak et al. (2006):

$$F_S = \left\langle \int uSdA \right\rangle \tag{1}$$

where the angle bracket is a 34-hour low-pass filter, $u$ is the axial velocity, $S$ is salinity, and $A$ is the cross-sectional area; the cross-sectional integral within the angle bracket represents the instantaneous salt flux. The total salt flux is further decomposed by:

$$
\begin{aligned}
F_S &= \left\langle \int (u_0 + u_E + u_T)(S_0 + S_E + S_T)dA \right\rangle \\
&\approx \left\langle \int (u_0 S_0 + u_E S_E + u_T S_T)dA \right\rangle \\
&= Q_f S_0 + F_E + F_T
\end{aligned}
\tag{2}
$$

in which u and S are decomposed into tidally and cross-sectionally averaged ($u_0$, $S_0$), tidally averaged and cross-sectionally varying ($u_E$, $S_E$), and tidally and cross-sectionally varying ($u_T$, $S_T$) components. The first term, $Q_f S_0$, is the salt flux resulting from subtidal cross-sectionally averaged transport, such as the salt loss due to river discharge, and is referred to as the advective flux. The second term, $F_E$, is the subtidal shear dispersion resulting from estuarine circulation, which is called as the steady shear transport flux. The third term is the tidal oscillatory salt flux resulting from temporal correlations between $u_T$ and $S_T$.

## 4 Model validation

### 4.1 Model calibration

The values of the bottom roughness height and vertical grid resolution are adjusted in the simulations for model calibration. We test roughness heights of 0.001m, 0.005 and 0.01m, and examine 8, 15 and 21 layers for the vertical resolution. The model results are compared with the observed data from November 3-4, 2005. We can not compare the modeled wave parameters with observed data because wave measurement data are not available.

On 3 and 4 November, 2005, 10 stations were deployed in the West Shoal and the West Channel (Fig. 1c), and hourly measurements were taken of the water current and salinity profiles for 27 hours continuously. The ADCPs and direct-reading current meters were utilized for collecting current data, and CTDs were used for salinity measurements.

The model simulations cover the period from September 20, to November 10, 2005. The initial conditions of water level and current are set to zero, while the initial conditions for salinity and temperature are obtained by running the model for approximately 3 months in the period preceding September 20, 2005, as previous studies indicated that an adjustment time of approximately 2 month during the winter season in the PRE is needed (Sun et al., 2014). To permit comparison of the model results and the observed data, the water column is divided into 6 layers and the model output is interpolated vertically to correspond with the measurements.





The model-data comparison is quantified using the root mean square error (RMSE), and the predictive skill (Skill), values of which are shown in Table 2. The magnitude of the RMSE indicates the average deviation between the model results and the observed data. The Skill is calculated following the method of Willmott (1981):

$$Skill = 1 - \frac{\sum_{i=1}^{N}(M_i - O_i)^2}{\sum_{i=1}^{N}(|M_i - \overline{O}| + |O_i - \overline{O}|)^2} \qquad (3)$$

where $M_i$ and $O_i$ are the ith model result and observation, respectively, and $\overline{O}$ indicates the time mean of the data, and $N$ is the number of observations. The Skill provides an index of model-data agreement, with a skill value of 1 indicating perfect agreement and a value of 0 indicating complete disagreement.

We compare the depth-average axial velocity ( $\overline{U}$ ), the salinity difference between the surface and the bottom ( $\Delta S$ ) (which represents the salinity stratification), and the depth mean salinity ( $\overline{S}$ ) variables. $\overline{U}$ at a station is obtained by defining the

principal direction of the depth-averaged velocity using the method of Emery and Thomson (2002). Comparison of the model results and the observations for a roughness height of 0.001m and a vertical grid resolution of 15 layers (Fig. 2 and Table 2) show that, overall, the model performs well. The skill values for $\overline{U}$ are larger than 0.88 at all stations, and for $\overline{S}$ and $\Delta S$ are mostly larger than 0.5, indicating excellent and good performance, respectively. The salinity stratification is poorly reproduced at Station S8, which maybe because of inaccuracy of the local bathymetry around the station.

For the vertical resolution, the 8 layer model generally underestimates the salt intrusion into the estuary and the vertical salinity stratification, and there are no significant differences in the modeled salinity for the 15- and 21-layer configurations (not shown). In previous studies of the PRE region, when implementing ROMS, Zu et al. (2014) and Zu and Gan (2015) used 30 layers in the vertical, and Pan et al. (2014) used a 20-layer vertical scheme. Their model domain did not include the Pearl River Network, with the result that the spatial resolution in the PRD is low. We choose a 15-layer vertical resolution as

we feel it provides a reasonable compromise between computational efficiency and model accuracy.

### 4.2 Model verification

As a further check of the model's ability to simulate the current and salinity, we verify it with an additional set of observed data. The observation was conducted over a period of 17 days, from December 9 to 26, 2009. A total of 6 stations were

installed along the Modaomen Estuary, another estuary in the PRD that is covered by our model domain. Hourly profiles of current and salinity profiles were obtained. As in the model calibration, tide, river discharge, winds and waves are forced in the model, and the timeseries of these external forcings are shown in Fig. 3. The model is run from November 1 to December 31, 2009. The modeled and observed data for current and salinity are shown in Table 3.

As shown in Table 3, the model results agree reasonably well with the measured data for both the velocity and salinity. The

model skill for the velocity at M2 is less than 0.5, reflecting the poor quality of the observed velocity data for this station. The skill for the velocity at M4 is approximately 0.5 and reflects the fact that the M4 station is in a side tributary with a narrow cross-section, where the model grid's resolution is not sufficiently high to capture the small scale dynamics. Values of skill for salinity indicate that the model satisfactorily reproduces the salinity variations at M2, M3, and M5 but underestimates the salinities at M1 and M6. The modeled values of salinity for M1 are approximately 1.5 ppt higher than the

observed data. The reasons for this are unclear, but maybe due to a local source of freshwater, that is not represented in the model.





On the whole, the model provides good simulations of the water currents and salinity in the study area, and the results indicate that it is suitable for further studies of salinity dynamics in the PRE.

## 5 Results

In this section, we report the results of the diagnostic simulations of salt intrusion dynamics for the period from November 1 to December 31, 2009. As shown in Fig. 3, during this period, the winds (Fig. 3a) are dominated by down-estuary winds (NE, N), interrupted by cross-estuary (E) and up-estuary (SE) ones. The maximum wind velocity is 15.4 m/s, corresponding to a NE wind pulse, while the mean wind velocity is 6.3 m/s. The total river discharge from the West, North and East rivers is less than 4000 $m^3$/s, with a mean value of 2162 $m^3$/s, representing a typical dry season situation. The subtidal water levels at
the western and eastern open boundaries are higher than the mean sea level during the simulation period, and the water level is even higher at the western boundary (Fig. 3c). The subtidal currents along the ocean boundaries are southwestward, and reach 0.3 m/s at the eastern and southern boundaries (Fig. 3d).

### 5.1 Salt intrusion without wind

In Case 1, the model is run with tides and river discharge. The 50-hour averaged currents and salinity at the surface and bottom during the neap (e.g. Day 40 to 42) and spring (e.g. Day 47 to 49) tides obtained from the model results are shown in Fig. 4. We choose an average value of 50 hours to reflect the irregular semi-diurnal tidal characteristics of the PRE.

During the neap tide, there is a very clear pattern in the bottom salinity (Fig. 4b), with salt intrusion mostly along the West Channel and the East Channel, and a very prominent tongue of high salinity in the West Channel. The 10-ppt contour reaches
the head of the estuary, and the bottom residual current is mostly landward. Two tongues of salinity that protrude seaward form at the surface along the West and East Channels (Fig. 4a). As the freshwater is discharged into the PRE from the northern and western sides, the salinity contours mostly align in a NE-SW direction. Inside the estuary, the surface residual flow is seaward with strong currents in the West and East Channels, while outside the estuary it is a westward coastal current. During the spring tide, the bottom salt intrusion (Fig. 4d) is also mostly along the channels but is less intense than that
observed during the neap tide, because of the enhanced mixing during the spring tide. The surface salinity (Fig. 4c) is smoothly aligned in a NE-SW direction and is generally higher during the spring tide than during the neap tide. The higher surface salinity during the spring tide are due to two facts: 1) lower freshwater inflow from the Pearl River Network during the spring tide than during the neap tide owing to the greater bottom friction during the spring tide (Buschman et al., 2009) ; 2) enhanced mixing during the spring tide.

The estimates of the length of the salt intrusion obtained from the model results are shown in Fig. 5. The length of the intrusion decreases during spring tides and increases during neap tides, which is consistent with what generally happens in a partially mixed estuary (Bowen and Geyer, 2003; Gong et al., 2011). It is noteworthy that from Day 25 to 28 and from Day 54 to 57 of the neap tides, the length of the salt intrusion increases substantially, but there is no meaningful increase in the length of the intrusion during the neap tide from Day 39 to 42. The reasons for such a behavior are discussed later.

### 5.2 Effect of local wind

In Case 2, local wind forcing at the water surface of the estuary is added to the river discharge and tides.

The salt intrusion is greatly enhanced relative to that without wind (Fig. 5b), and the length of the intrusion increases by an average of 6.7 km during the simulation period. As an example, the tidally averaged (a spring tide from Day 47 to 48 when



the NE wind was strong) surface and bottom salinities between with and without the local wind are compared in Fig. 6. The surface and bottom salinities in the estuary both increase considerably when the local wind is applied.

To determine what controls this increase in salt intrusion, we examine the changes in the salt transport fluxes at the Section T2. The salt transport flux components (the advective, steady shear and tidal oscillatory) and the total salt flux are shown in Fig. 7.

For Case 1, the advective flux (Fig. 7a) at the Section T2 is seaward, and is generated by a mean seaward flow. It fluctuates on a fortnightly cycle, with larger fluxes during neap tides and smaller ones during spring tides. Changes in the water volume flux are mostly responsible for these fluctuations, while the changes in mean salinity are minor. There are two parts to the tidally averaged water volume flux: the river inflow and the tide-induced Stokes flux. During neap tides, the river inflow into the PRE is larger, because of the reason mentioned above, and the landward Stokes flux is smaller, because of the predominantly progressive tidal characteristic in the estuary (Mao et al., 2004), and the combination of these two parts means that the seaward flow is stronger during neap tides than during spring tides.

Under the local wind, the seaward advective flux decreases by 21% (Fig. 7a), with the mean seaward flow decreasing by 25.8%, and the mean salinity increasing by 5.7%. Apparently, the decrease in the seaward salt flux is caused by the decrease in the volume of water transported seaward, with the reasons being illustrated later. The decrease in the seaward transport corresponds well with the occurrence of the NE winds.

The landward steady shear transport (Fig. 7b) is larger during neap tides and smaller during spring tides in Case 1, which is characteristic of a partially mixed estuary (MacCready and Geyer, 2010). The increase from Day 40 to 42 is not as great as during other two neap tides, and is consistent with the moderate increase in salt intrusion observed during that period (Fig. 5b). The reason for such a moderate increase could be that the river discharge during that period is low (Fig. 3a), which increases the time for a parcel travelling from upstream to downstream. The adjustment time of an estuary is proportional to this travelling timescale (MacCready, 2007), and thus maybe longer than or comparable to the fortnightly timescale of the spring-neap cycle, making the increase in salt intrusion during this neap tide limited (Lerczak et al., 2009).

The steady shear transport changes considerably under local wind conditions (Fig. 7b), and is increased when the NE winds prevail and decreased under other wind directions. This suggests that the NE winds enhance the estuarine circulation and stratification at the estuary mouth.

The tidal oscillatory flux (Fig. 7c) is relatively small for both Cases 1 and 2. Generally, the local wind causes the landward tidal flux to decrease.

Overall, the total salt transport flux (Fig. 7d) is seaward and decreased considerably under the local wind, which explains why the salt intrusion increases when the local wind is forced. This decrease in the seaward flux mostly reflects the decrease in the volume of water transported seaward.

It is noted that during the study period, the instantaneous and cumulative total salt flux at the Section T2 is seaward, while the salt intrusion does not show a continually decreasing trend. We analyzed all the "in and out" salt transport fluxes of the estuary and found that the four outlets at the northern and western sides, and the East Entrance of the estuary, are importing salt into the estuary. These salt imports balance out the salt export at the Section T2, which is similar to that by Lai et al. (2016). Another feature is that larger salt exports at the Section T2 correspond with stronger bottom (Fig. 5) and weaker surface (not shown) salt intrusions at the West Channel during neap tides, and vice versa during spring tides.

To obtain a clearer picture of the hydrodynamic changes induced by the local wind, we compare the subtidal water level in the estuary, mean current and salinity at the Section T2 between Cases 1 and 2 (Fig. 8). With the local wind, the subtidal water level is lowered in the eastern part of the estuary, and becomes parallel to the shoreline in the western part of the estuary (Fig. 8a). At the Section T2, the salinity without wind manifests a partially mixed state, with a minimum salinity of 27 ppt observed at the western shore. The residual current pattern shows a maximum seaward flow at the surface of the West Channel, and a seaward flow through most of the section. With the local wind, there is more mixing of salinity, and the



salinity declines to a minimum of 23 ppt at the western shore, but increases from 31 to 33 ppt in the West Channel. Meanwhile, there develops a coastal jet along the western shore, and the residual flow changes from vertically sheared to horizontally sheared, with the maximum seaward flow shifting from the surface of the West Channel to the western shore. Landward flow occupies most of the West Channel and is enhanced. This explains the decrease in the volume of water

transported seaward and the advective salt flux under the local wind. The strong horizontal segregation of the water current and salinity results in enhanced steady shear transport.

### 5.3 Effect of remote wind

In Case 3, wind forcing is applied outside the estuary. We also superimpose the subtidal water levels and currents from the
HYCOM outputs onto the tidal components at the open ocean boundaries. Under the remote wind, the length of the salt intrusion increases by an average of 5.1 km (Fig. 5b). Comparison of the isohalines during the spring tide for Cases 1 and 3 (Fig. 9) shows that with the remote wind, the salinity increases more at the surface than at the bottom. This is consistent with the pattern induced by a downwelling-favorable wind on the shelf (Hickey et al., 2002). The details will be examined later.
The change in the subtidal water level under the remote wind indicates that there is a distinct increase in the water level
across the entire domain under the remote wind, with a mean increase of 0.15 m in the estuary (not shown). Along with the increase of water level, the mean seaward flow at the Section T2 decreases by 16%, and the mean salinity at the Section T2 increases by approximately 0.6 ppt.
Under remote wind forcing, the seaward advective flux at the Section T2 (Fig. 7a) decreases noticeably during spring tides, but only slightly during neap tides. We estimate the water transport volume by remote wind induced subtidal water level
setup/setdown at the Section T2, following Janzen and Wong (2002):

$$F_w = A_s \frac{\partial \eta}{\partial t} \qquad (4)$$

where $F_w$ is the water flux, $A_s$ is the surface area of the estuary, and $\eta$ is the low-pass filtered water elevation at the Section T2. As downwelling-favorable winds prevail in the simulation period, the water influx increases during wind accelerating periods, and decreases or even reversed during wind decelerating periods or when the wind becomes upwelling-favorable.
Strong tidal current during spring tides augments the wind-induced influx. From Day 46 to 48 (spring tide), the calculated water influx is about 2000 $m^3$/s, while from Day 41 to 43 (neap tide), the water influx is less than 1000 $m^3$/s. These changes in barotropic water flux affect the advective salt flux. During spring tides, the decrease in the seaward flow dominates over the increase in salinity, resulting in a noticeable decrease in the seaward advective flux. The salinity increase during spring tides is small because of strong mixing on the shelf. During neap tides, the decrease in the mean seaward flow cancels out
the increase in the mean salinity, such that the change in the advective flux is negligible. Meanwhile, the landward steady shear transport (Fig. 7b) decreases relative to that observed in Case 1, and is attributed to a reduction in estuarine circulation. The tidal oscillatory landward flux (Fig. 7c) decreases under the remote wind. Overall, there is less seaward salt transport (Fig. 7d) at the Section T2 in Case 3 than in Case 1, which results in an increase in salt intrusion under the remote wind.
The effect of the remote wind can be further explored by examining the changes in the currents and salinity at a cross-section
in the shelf (T3 in Fig. 1c; Fig. 10). In Case 1, a zone of higher salinity (34 ppt) is sandwiched between two lower salinity zones. The outer zone of lower salinity forms from offshore and upstream (relative to Kelvin wave propagation) spreading of the Pearl River plume due to pulsation of river discharge or spring-neap tidal variation, or both (Yankovsky et al., 2001; Rong and Li, 2012). With the remote wind, the river plume is forced to move downstream, the salinity at the cross-section is elevated, and the vertical mixing increases. A downwelling circulation is well developed with surface onshore flow and
bottom offshore flow. This circulation is to the reverse of, and thus reduces, the estuarine circulation at the Section T2. It also works to increase mixing outside the estuary mouth, and its effect is similar to that of tidal straining during flood tides



(Simpson, 1990). The increased mixing causes an increase in surface salinity near the Section T2, which propagates into the estuary and generates a greater increase in surface salinity than in bottom salinity (Fig. 9). This is consistent with the conceptual model proposed by Hickey et al. (2002). The remote wind also greatly accelerates the alongshore southwestward current, inducing stronger landward Ekman transport.

On the whole, the remote wind serves to pump saltier water into the estuary mouth, thereby decreasing the volume of water and salt transported seaward, and increasing salt intrusion in the estuary.

**5.4 Effect of wind wave**

We designed two simulations for examining the effects of wind waves. In Case 4, the local and remote winds are specified,
but the wave effects, including the bottom stress enhancement, wave-induced mixing and excessive momentum flux, are not activated. In Case 5, we activates these wave effects with both the local and remote winds. We compare the outputs of these two cases.

In Fig. 5, the total wind effect is simply the sum of the effects of the local and remote winds. The length of the salt intrusion increases by an average of 12 km with wind relative to that without wind. While the salt intrusion increases in length because
of the wind effect, the wave effect causes it to decrease by an average of 2 km.

The example of the wave height distribution under a strong NE wind (Day 47), presented in Fig. 11a, shows that significant wave heights are < 1 m in the estuary on this day. Figures 11b-11f show time series of significant wave height, current-induced bottom stress, wave-induced bottom stress, vertically-averaged eddy diffusivity, and bottom salinity with and without waves at a station in the West Channel (as indicated in Fig. 1c). The water depth at the station is 12.4 m. During the
simulation period, the wave height varies from 0.1 to 0.9 m. The current-induced bottom stress fluctuates from 0.01 to 1.5 $N/m^2$ and is much larger than the wave-induced one, which is less than 0.2 $N/m^2$. This indicates that the wave effect is not very significant at the station during the simulation period. Consequently, the variation in eddy diffusivity with waves generally follows that without waves. The eddy diffusivity, however, is elevated when the wave height is greater than 0.5 m and the wave-induced bottom stress exceeds 0.05 $N/m^2$. On average, the eddy diffusivity is 11% greater with waves than
without waves. Accordingly, the bottom salinity decreases by an average of 0.5 ppt as high salinity bottom water becomes increasingly mixed into the upper part of the water column. We also check the modeled currents in Case 5, which is the summation of Eulerian current and Stokes drift, and note that the wave-induced current is insignificant at this station because of the small wave height and large water depth.

We can see further evidence of the wave effect by comparing the salinity, mean current, and eddy diffusivity for Cases 1, 4,
and 5 (Fig. 12) along the longitudinal transect (T1 in Fig. 1c). The vertical mixing and the salinity at the lower part of the estuary (within 35 km from the estuary mouth) increase because of the winds, resulting in enhanced salt intrusion. Comparison of Cases 4 and 5 shows that with the waves, vertical mixing increases further (Fig. 12f compared to Fig. 12d), and salt intrusion decreases.

**5.5 Salt intrusion in response to different wind directions and associated waves**

In the above diagnostic study, we force the model with a wind field that varies both spatially and temporally. Some of the mechanisms that drive salt intrusion may be difficult to identify because of the transient wind field. We designed an additional experiment suite to investigate the response of salt intrusion to various wind directions with or without the associated waves. Cases 6, 7, and 8 are forced by N, NE and E winds, respectively, as these are the most frequent wind
directions during the dry season. In these cases, the open ocean boundaries are simply forced by tidal elevation and current. For each case, the wind magnitude is set to the mean wind velocity of the corresponding direction. The model is first run for





17 days without wind forcing, after which the wind is applied to the water surface. We use a spin-up period of 1 day, in which the wind forcing is ramped from zero to the mean wind stress through a sine function, followed by a constant wind stress for another 12 days. We carry out three simulations without and with the associated wave effects, respectively.

The changes in the length of the salt intrusion under different wind directions with and without the associated waves are

shown in Fig. 13. As can be seen in Fig. 13b, the strengthening of the salt intrusion is greatest with the axial down-estuary (N) wind, followed by the NE wind. The effect of the cross-estuary (E) wind is complicated: initially it triggers an increase in salt intrusion, but then, as the wind persists, it decreases the salt intrusion. When the wave effect is activated (Fig. 13c), the mean significant wave heights in the estuary are approximately 0.4, 0.3 and 0.5 m under the N, NE, and E winds, respectively. The superimposition of wave has similar effects under the three directions, which causes a decrease in salt

intrusion relative to that without wave.

We examine the salt intrusion dynamics under different wind directions. The salinities, currents and eddy diffusivities along the longitudinal transect (T1 in Fig. 1b) for the three wind directions are shown in Fig. 14. The estuarine circulation is most enhanced, and the water column is highly stratified, under the axial down-estuary (N) wind. The estuarine circulation is greatly weakened, and the water column is most mixed, under the cross-estuary (E) wind. The NE wind generates an

intermediate effect on the salt intrusion.

The effects of different winds can be seen from the graphs of the tidally mean salinity and current at the cross-section in the middle of the estuary (T4 in Fig. 1b) shown in Fig. 15. The axial down-estuary (N) wind generates a strong surface seaward flow, along with an enhanced bottom landward flow. Lateral flow in both the West and East Channels is in a pattern of flow convergence at the surface and westward flow near the bottom, and the water column is partially mixed. Under the cross-

estuary (E) wind, the seaward flow in the section is greatly reduced, and the landward flow is more confined in the East Channel. The water column is well mixed, and a uni-directional westward lateral flow develops. The change caused by the NE wind is intermediate.

## 6 Discussion

In this study, we examine the response of salt intrusion to winds and waves in the PRE using the COAWST modeling system.

Using a 1-D model, Mo et al. (2007) predicted that all N, NE, E and SE winds would have a depressing effect on salt intrusion in the PRE. Our results are very different, probably because of different models used and different hydrodynamic conditions between the main-stem and the tidal river of the PRE. The tidal river part is mostly well mixed, while the mainstem is partially mixed, and this difference largely controls the different response of salt intrusion to winds. Our results are consistent with the findings of Lai et al. (2016), who used the Finite-Volume Community Ocean Model (FVCOM) and

found that the NE wind enhances salt intrusion in the PRE during the dry season. Our results for the PRE are also consistent with the observations of Wen et al. (2007) from their study of the neighboring Modaomen estuary. They examined the observed data and noticed that the down-estuary (NE and N) winds favor increasing salt intrusion in that estuary.

The NE wind is the most frequent wind in the dry season. The remote wind effect of the NE wind is to drive saltier shelf water into the estuary, and increase the salinity at the estuary mouth. Landward transport by the remote wind is simply

proportional to the wind stress, as suggested by the Ekman transport, as follows: $U = \dfrac{\tau_w}{\rho f}$ , where $\tau_w$ is the wind stress, $f$

is the Coriolis parameter, and $\rho$ is the water density. One may expect that the remote wind effect would increase as the wind strength increases, which is consistent with our previous estimation of remote wind-induced subtidal water flux at the Section T2.

For the local wind effect, the effect of the NE wind is not just that of the axial wind, but rather the combination of those of

the axial down-estuary (N) wind and the cross-estuary (E) wind. This wind pushes the plume water towards the western



shore, lowers the water level in the eastern part of the estuary, decreases the seaward water level gradient, and increases the bottom landward flow, thereby enhancing the salt intrusion. In the meantime, the NE wind increases the water mixing and retards the salt intrusion. The local wind effect is therefore a competition between the wind straining and the wind mixing. Chen and Sanford (2009) have intensively explored the transition of the estuary's stratification and estuarine circulation in

response to changes in wind velocity magnitude and direction. A nondimensional parameter, the Wedderburn number, has been proposed to express the relative importance of the wind stress and the baroclinic pressure gradient (Monismith, 1986), and also to signify the relative importance of wind straining and wind mixing:

$$W = \frac{\tau_w L}{\Delta \rho g H^2} \qquad\qquad (5)$$

where $L$ is the estuary length, defined as the salt intrusion length; and $H$ is the mean water depth in the estuary, which is

defined as the mean water depth along the West Channel. The mean Wedderburn number for our simulation period is 0.1, which is much smaller than the critical value of ~1 (0.85 in Chen and Sanford, 2009) and suggests that the wind straining effect dominates over the wind mixing. The Wedderburn numbers for the mean down-estuary (N, NE) winds are 0.14 and 0.08, respectively, again indicating that the wind straining dominates. One may expect that the wind straining effect would dominate most of time during the dry season in the PRE, as the wind stress is 0.4 pa and the wind speed is 14 m/s when the

Wedderburn number is greater than 1; however, this would indicate a winter storm, which occurs infrequently in our study area, and happens only once during our simulation period.

Our results show that increases in the salt intrusion caused by the remote wind and the local wind are comparable, which suggests that the remote wind should not be ignored when modeling the wind effect on salt intrusion. The remote wind effect in our model simulation accounts for the higher water elevation at the open ocean boundaries and is similar to the sea level

rise studied by Yuan et al. (2015) in the PRE. Our model results show that the higher water level in the model domain enhances the estuarine circulation, especially the bottom landward flow, and increases the salt intrusion. There will be more concern about this effect in the future as sea level rise accelerates.

As mentioned above, the effect of the cross-estuary (E) wind on salt intrusion is complicated: The cross-estuary (E) wind causes an initial increase followed by a decrease, while the down-estuary (N and NE) winds generate continuous increases in

salt intrusion (Fig. 13b and 13c). We explore the mechanisms behind this difference. Time series of the vertically averaged salinity at the Section T2, mean eddy diffusivity, and mean bottom current along the longitudinal transect (T1) are presented in Fig. 16. All the winds induce an increase in salinity at the estuary mouth (Fig. 16a), because of Ekman transport of saltier water from the shelf by the downwelling-favorable winds (NE and E), or an increase in the bottom landward flow by the wind straining effect from the down-estuary (N) wind. Eddy diffusivity along the T1 transect (Fig. 16b) is greatly enhanced

under the three winds, and in particular by the cross-estuary (E) wind. The cross-estuary (E) wind induces a strong westward lateral flow (Fig. 15e), which results in a stronger landward flow at the surface than at the bottom by the Coriolis force. Momentum balance analysis in the longitudinal direction indicates that, near the surface, the Coriolis force almost balances out the pressure gradient force (not shown),indicating a geostrophic balance. This landward flow increases the velocity shear and tidal straining effect during flood tides and decreases the velocity shear and tidal straining effect during ebb tides,

resulting in greatly enhanced mixing as a whole. Consistent with these changes in mixing, the bottom landward flow is significantly reduced by the cross-estuary (E) wind (Fig. 16c). However, the down-estuary (N and NE) winds cause an increase in the bottom landward flow because of the increased landward water level gradient by these two winds. Consequently, the initial increase in salt intrusion under the cross-estuary (E) wind is mostly due to the increase in salinity at the estuary mouth promoted by Ekman transport on the shelf, while the subsequent decrease in salt intrusion by the cross-

estuary (E) wind is induced by increased mixing and decreased bottom landward flow. The continuously increases in salt intrusion caused by the down-estuary (N and NE) winds are mostly due to increased bottom landward flow, while the



downwelling-favorable (NE) wind causes more of an increase in salinity at the estuary mouth than the down-estuary (N) wind.

The cross-estuary wind effect on salt intrusion in the world's estuaries has been studied previously (e.g., Xu et al., 2008; Wang et al., 2016). In the PRE, the cross-estuary (E) wind tends to reduce salt intrusion, as it increases water mixing and reduces estuarine circulation. One might speculate that the cross-estuary W wind would increase estuarine circulation, and enhance salt intrusion, though the W wind seldom occurs during the dry season. This is confirmed in another numerical experiment (not shown), in which we specify a constant W wind with the same magnitude as the E wind in the model domain, and the salt intrusion increases slightly (~2 km).

Because of a lack of observed wave data, we do not validate the model results of wave parameters in this study. However, our model outputs for a wave monitoring station outside the PRE shows that the mean significant wave height and maximum wave height are approximately 0.95 m and 1.81 m, respectively, during the simulation period. These values are similar to the statistic value at the station, and give us confidence that the wave model is reliable.

The wave effects comprise: 1) bottom shear stress enhancement due to wave-current interaction, 2) increase of mixing because of surface wave breaking and white-capping, and 3) excessive momentum flux in the water column, including the Bernoulli head, vortex forces, and the non-conservative wave forcing, e.g. bottom and surface streaming and wave breaking. As discussed earlier, the two former effects are to increase mixing, while the third effect is not obvious, so we conduct another experiment in which a constant NE wind and waves are activated and the wave-induced excessive momentum fluxes are turned off.

Results of the scenarios with and without wave-induced momentum fluxes indicate that there is a negligible decrease in the length of the salt intrusion when the wave-induced momentum fluxes are turned off. Examination of the mean bottom landward current along the T1 section shows that the bottom landward flow is slightly higher with the wave-induced momentum fluxes than without them. This shows that the wave effect is mainly comprised of the enhanced mixing caused by surface and bottom wave processes during the simulation period. This conclusion may not be applied to the period when the waves are large, such as during typhoon events.

As indicated in Fig. 5b, compared with the wind effect, the wave effect is minor in increasing the salt intrusion. However, this does not mean that the wave effect is not important in estuarine dynamics in the PRE. We mainly examine salt intrusion in the West Channel, where the water depth is about 10m deep and the wave effect is relatively weak. However, the wave effect on shoals in the estuary can be significant, with implications for dispersal of the river plume. The river plume in the PRE comes from the outlets along the western shore and passes through the West Shoal before reaching the estuary mouth. We would expect that waves would have a sizeable effect on river plume dynamics in the West Shoal, but this is beyond the scope of the present study.

**7 Conclusions**

In this study, we used the COAWST modeling system to study the effects of wind and waves on salt intrusion in the PRE during the dry season. We validated the model with observed data and then used it to complete 8 exploratory experiments. Besides the tides and river discharge, we investigate the local and remote winds, wind waves, winds in three different directions and the associated waves to determine their respective effects. From our analysis, we find that:

1) Salt intrusion under the tides and river discharge follows the pattern expected for a partially mixed estuary, and increases during neap tides and decreases during spring tides. This pattern forms because the river discharge is higher, and the mixing is lower, during neap tides. These conditions favor a steep salinity gradient and stronger estuarine circulation, resulting in enhanced landward salt transport.



2) When we include winds in the model, the salt intrusion increases by an average of 12 km. After more detailed investigation of the respective effects of local and remote winds, we find that the local wind pushes the plume water against the western shore in the estuary, lowers the water level in the eastern part of the estuary, and increases the bottom landward flow, favoring an increase in salt intrusion. It also generates horizontally-sheared estuarine circulation and segregated salinity

distribution. The remote wind pumps saltier water into the estuary mouth, increases the vertical mixing on the shelf, and generates a change in the salinity pattern such that there is a noticeable increase in surface salinity in the estuary, but only a slight increase in the bottom salinity. The increases in salt intrusion due to the remote wind and the local wind are comparable. The remote wind effect is somewhat similar to the effect of a rise in sea level, which enhances intrusion by increasing the water level and estuarine circulation.

3) The axial down-estuary (N) wind is most efficient at driving saltier water landward by the wind straining effect. The cross-estuary (E) wind largely increases the vertical mixing, and reduces the salt intrusion. The effect of the NE wind is somewhere between the effects of the cross-estuary (E) and axial down-estuary (N) winds.

4) The wind wave is not strong during our study period, and has a significant wave height of approximately 1 m. The wave effect serves mostly to enhance the vertical mixing and decrease the salt intrusion. Generally, the wave effect on salt

intrusion is minor in our study period because the wave heights are relatively small and the water is deep in the West Channel. The wave effect may become more significant under stronger waves and/or in shallow areas. A study of the effect of waves on salt intrusion during storm conditions has been conducted and will be presented in a companion paper.

5) Our results would be applicable to other partially mixed estuaries in the world which are under the threat of salt intrusion, e.g., the Yangtze River estuary (Wu et al., 2010).

**Acknowledgements**

This research is funded by the National Natural Science Foundation of China [grant numbers:41576089, 41506102], the national key research and development program of China (2016YFC0402603) and the Guangdong Provincial Key Research [grant number:2014A030311046]. It is also supported by the Special Program for Applied Research on Super Computation

of the NSFC-Guangdong Joint Fund (the second phase). We would like to express our appreciation to Yuren Chen at the Sun Yat-sen University for his help in editing the figures and also to John C. Warner, USGS, for his providing the COAWST source code and technical support.

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



Table 1 Model simulation cases

| Model cases | Tides | River discharge | Local wind | Remote wind | Subtidal water level and velocity | Wave |
|---|---|---|---|---|---|---|
| 1 | √ | √ | × | × | × | × |
| 2 | √ | √ | √ | × | × | × |
| 3 | √ | √ | × | √ | √ | × |
| 4 | √ | √ | √ | √ | √ | × |
| 5 | √ | √ | √ | √ | √ | √ |
| 6 | √ | √ | N | N | × | × (√) |
| 7 | √ | √ | NE | NE | × | × (√) |
| 8 | √ | √ | E | E | × | × (√) |





Table 2 Comparison between model results and observation in November, 2005

| Stations | RMSE | | | Skill | | |
|---|---|---|---|---|---|---|
| | $\overline{U}$ (m/s) | $\Delta S$ (ppt) | $\overline{S}$ (ppt) | $\overline{U}$ (m/s) | $\Delta S$ (ppt) | $\overline{S}$ (ppt) |
| S1 | 0.033 | 0.396 | 0.386 | 0.985 | 0.640 | 0.675 |
| S2 | 0.032 | 0.613 | 0.675 | 0.953 | 0.739 | 0.641 |
| S3 | 0.032 | 0.557 | 0.818 | 0.976 | 0.622 | 0.754 |
| S4 | 0.037 | 0.984 | 0.324 | 0.962 | 0.630 | 0.752 |
| S5 | 0.023 | 0.821 | 0.901 | 0.962 | 0.509 | 0.687 |
| S6 | 0.054 | 0.718 | 0.775 | 0.886 | 0.688 | 0.585 |
| S7 | 0.020 | 0.922 | 0.981 | 0.977 | 0.557 | 0.586 |
| S8 | 0.032 | 0.693 | 0.578 | 0.968 | 0.434 | 0.641 |
| S9 | 0.031 | 0.519 | 1.063 | 0.978 | 0.634 | 0.569 |
| S10 | 0.046 | 0.559 | 0.478 | 0.953 | 0.701 | 0.643 |



Table 3 Comparison between model results and observation in December, 2009

| Stations | Vertical layer | Velocity (m/s) | | Salinity (ppt) | |
| --- | --- | --- | --- | --- | --- |
| | | RMSE | Skill | RMSE | Skill |
| | Bot. | 0.010 | 0.881 | 0.376 | 0.409 |
| M1 | Mid. | 0.015 | 0.917 | 0.464 | 0.332 |
| | Sur. | 0.021 | 0.876 | 0.559 | 0.484 |
| | Bot. | 0.025 | 0.453 | 0.262 | 0.633 |
| M2 | Mid. | 0.039 | 0.458 | 0.350 | 0.408 |
| | Sur. | 0.044 | 0.511 | 0.432 | 0.483 |
| | Bot. | 0.010 | 0.907 | 0.267 | 0.720 |
| M3 | Mid. | 0.019 | 0.885 | 0.266 | 0.766 |
| | Sur. | 0.020 | 0.857 | 0.192 | 0.767 |
| | Bot. | 0.009 | 0.774 | 0.180 | 0.528 |
| M4 | Mid. | 0.021 | 0.605 | 0.191 | 0.660 |
| | Sur. | 0.023 | 0.545 | 0.183 | 0.605 |
| | Bot. | 0.011 | 0.869 | 0.282 | 0.680 |
| M5 | Mid. | 0.013 | 0.921 | 0.203 | 0.757 |
| | Sur. | 0.021 | 0.849 | 0.119 | 0.811 |
| | Bot. | 0.017 | 0.775 | 0.488 | 0.398 |
| M6 | Mid. | 0.021 | 0.862 | 0.348 | 0.407 |
| | Sur. | 0.026 | 0.814 | 0.257 | 0.425 |





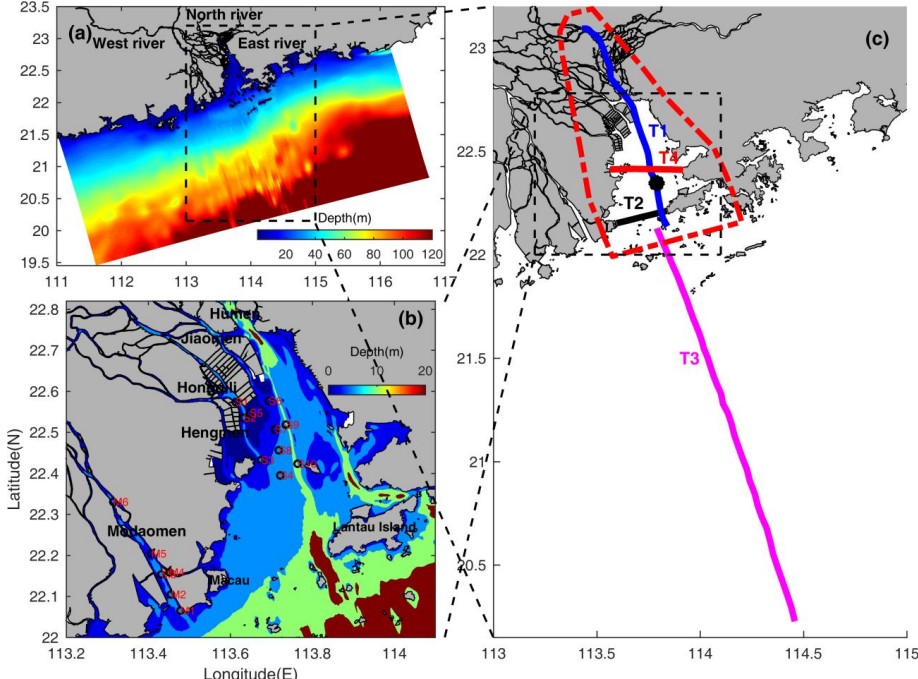

Figure 1. a) The study site and model domain. The West, North and East Rivers are three branches of the Pearl River, creating the Pearl River Network in the Pearl River Delta (PRD). b) Bathymetry of the PRE (in the right) and the Modaomen estuary (in the lower-left), and locations of the measurement stations. The two green stripes in the PRE show the locations of

5    the West and East Channels; adjacent to the two channels are the West, Middle and East Shoals. c) Sections and station for analysis. The red dash box is the region of the PRE. T1 is the longitudinal transect along the West Channel and the upstream tidal river; T2 the cross section at the estuary mouth; T3 is the cross-section on the continental shelf; T4 is the cross-section in the middle of the estuary; the black dot is the station for analysis in the text.



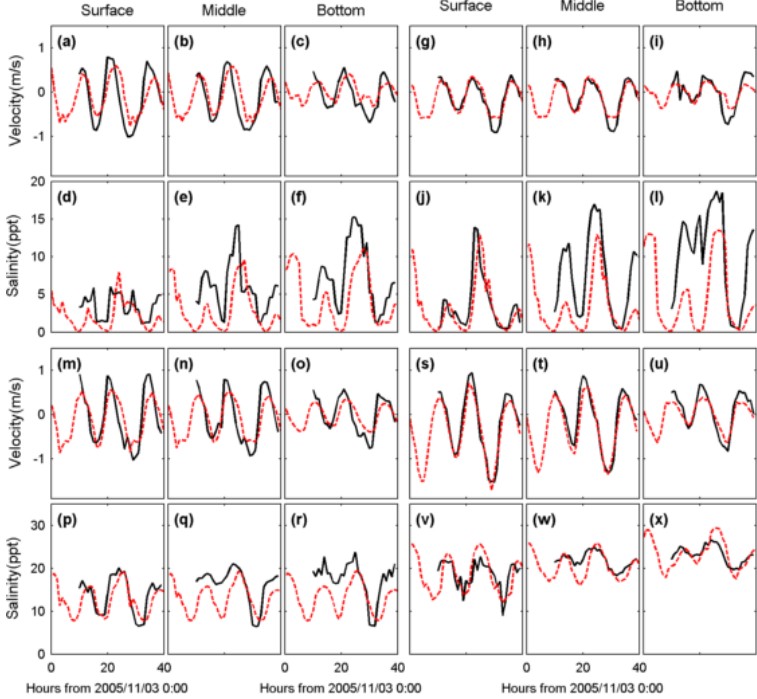

Figure 2. Model calibration for the period of November 3-4, 2005. (a)-(c) are for axial current, (d)-(f) are for salinity at the surface, middle and bottom layers at Station S2; (g)-(i) are for axial current, (j)-(l) are for salinity at Station S5;(m)-(o) are for axial current, (p)-(r) are for salinity at Station S6;(s)-(u) are for axial current, (v)-(x) are for salinity at Station S9. The red
5    dashed line stands for the model results, while the black solid line is the observations.



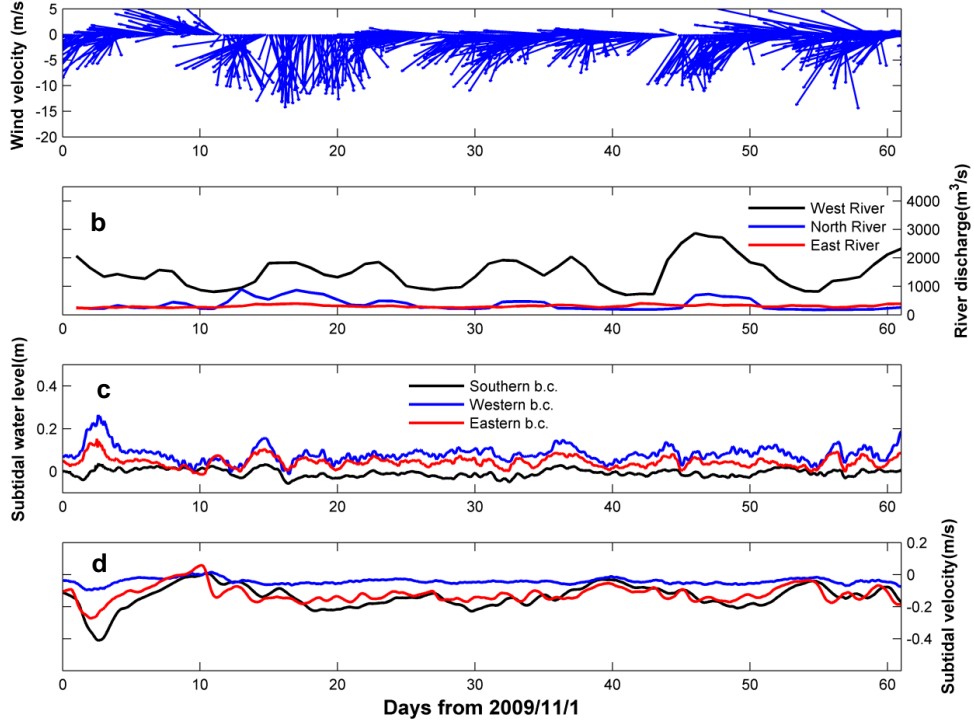

Figure 3. The external forcings for the model verification period from November 1, to December 31, 2009. a) time series of wind vector at the Hong Kong Meteorological station; b) daily river discharge from the West, North and East Rivers. c) subtidal water levels at the open ocean boundaries; d) subtidal alongshore current at the open ocean boundaries.




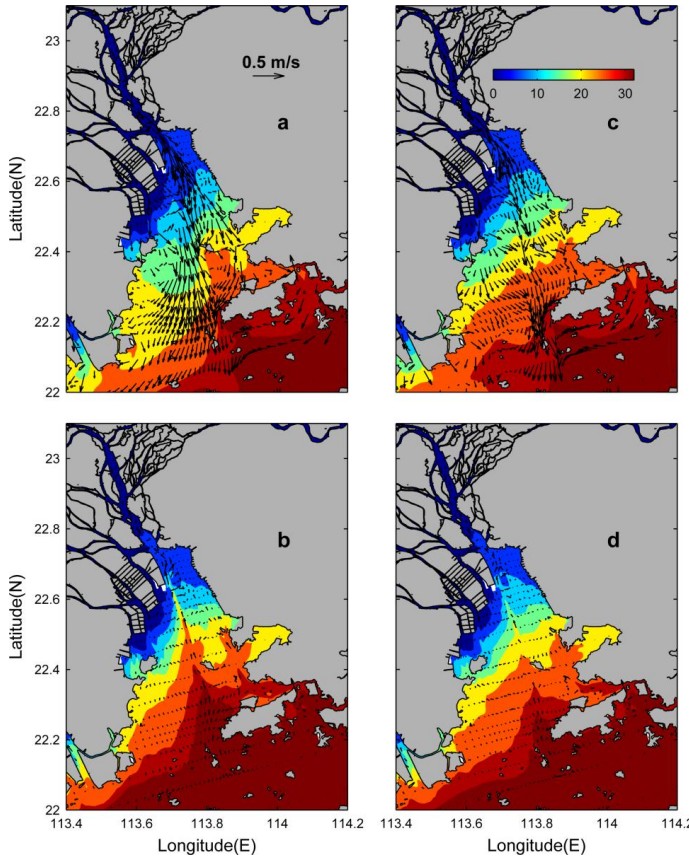

Figure 4. The modelled distribution of 50-hour averaged salinity and current at a) the surface; b) the bottom during a neap tide; and at c) the surface; d) the bottom during a spring tide.

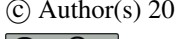



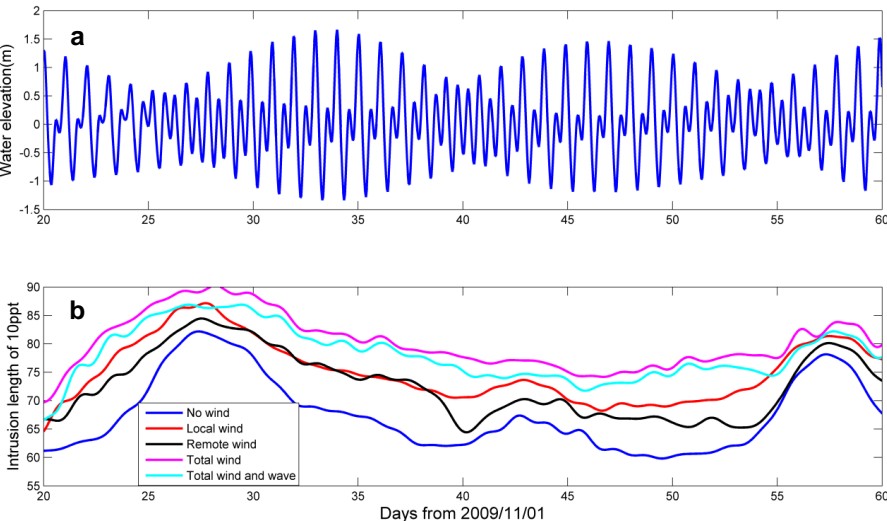

Figure 5. a) Time series of water elevation at the estuary mouth; b) Low-pass filtered length of the salt intrusion under different cases.





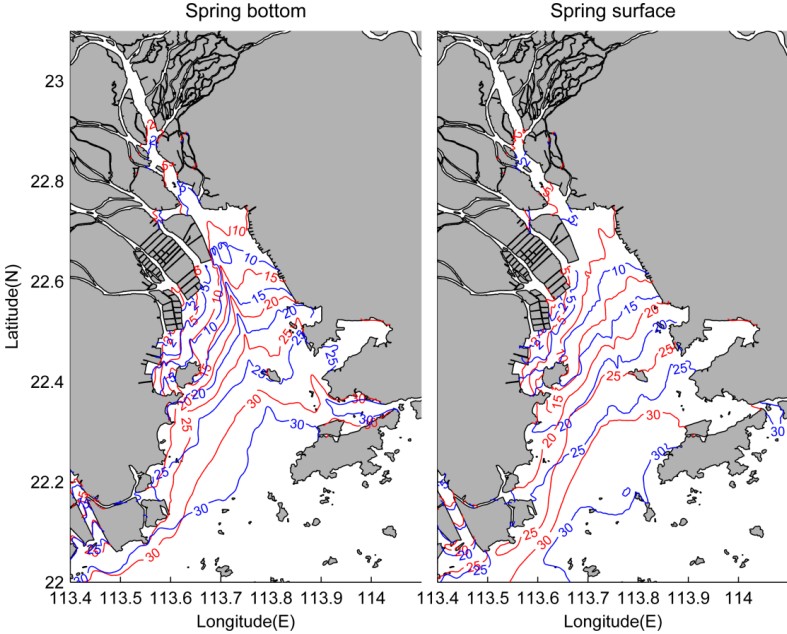

Figure 6. Comparison of 50-hour averaged salinity contours for Cases 1 and 2. The left panel is for the bottom salinity during a spring tide, while the right panel is for the surface salinity during the spring tide. The blue contour stands for case 1, while the red contour is case 2.




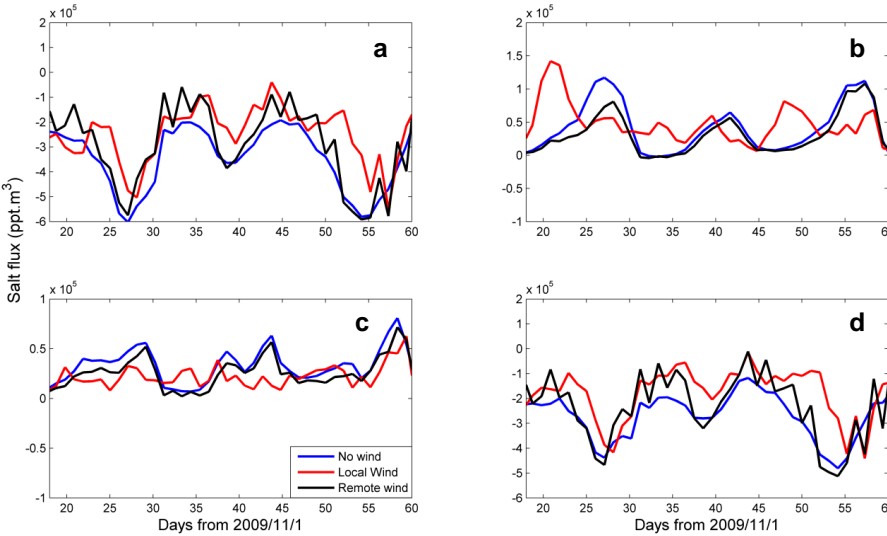

Figure 7. The salt flux under different cases at the cross section T2 of the estuary mouth. a) The advective; b) The steady shear; c) The tidal oscillatory; d) The total salt flux.

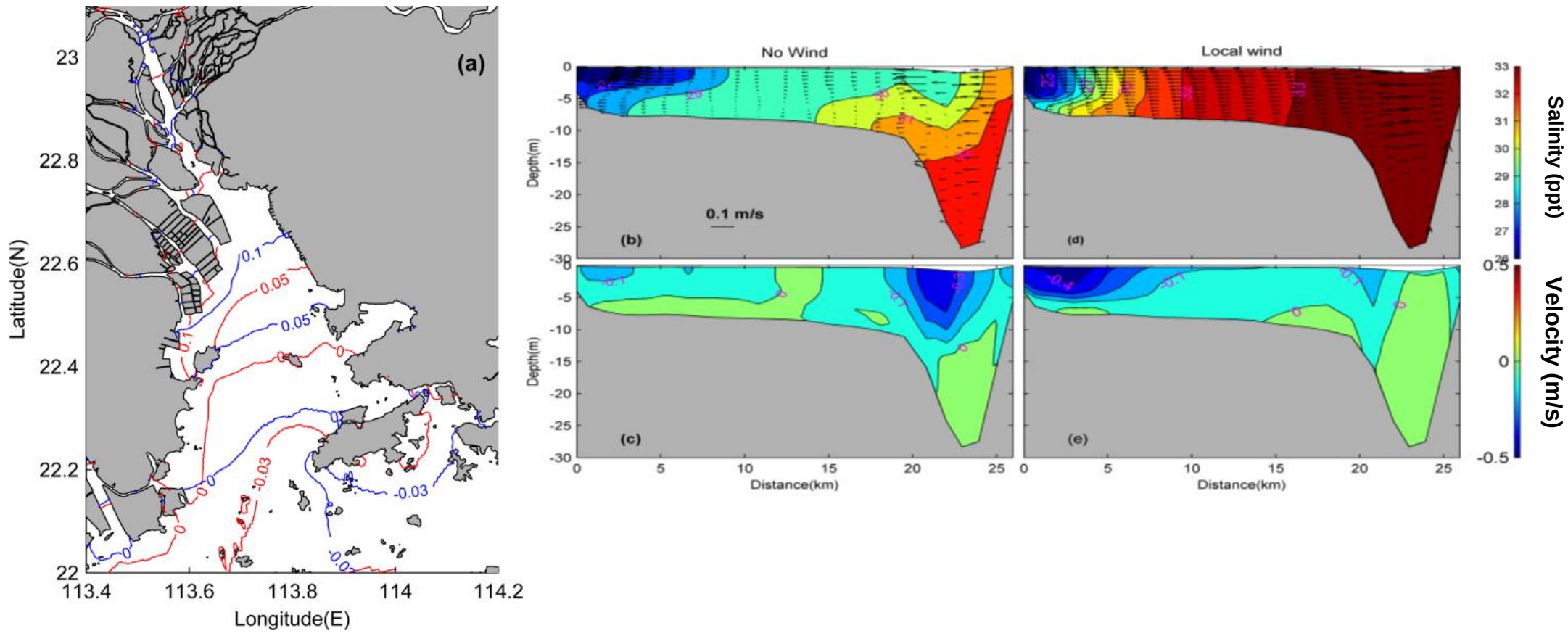

Figure 8. Comparison of subtidal water level during the spring tide for Cases 1 and 2 (a). The blue line depicts Case 1, while the red line represents Case 2. and comparison of 50-hour averaged salinity and current at the estuary mouth (T2) for Cases 1 and 2; (b) The salinity and lateral flow in Case 1; (c) The axial current in Case 1; (d) The salinity and lateral flow in Case 2; (e) The axial current in Case 2. The negative values in (d) and (e) mean seaward currents.





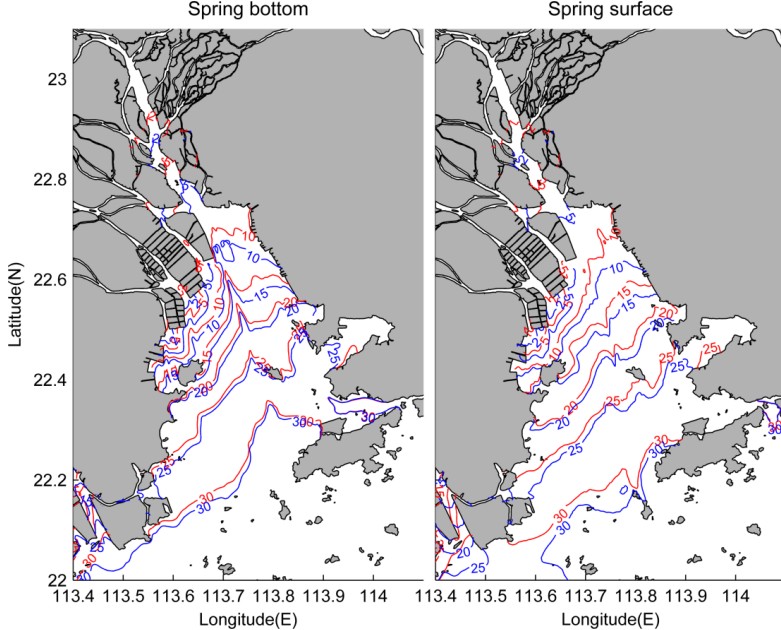

Figure 9. Comparison of salinity during the spring tide for Cases 1 (blue contour) and Case 3 (red contour): a) the bottom; b) the surface.




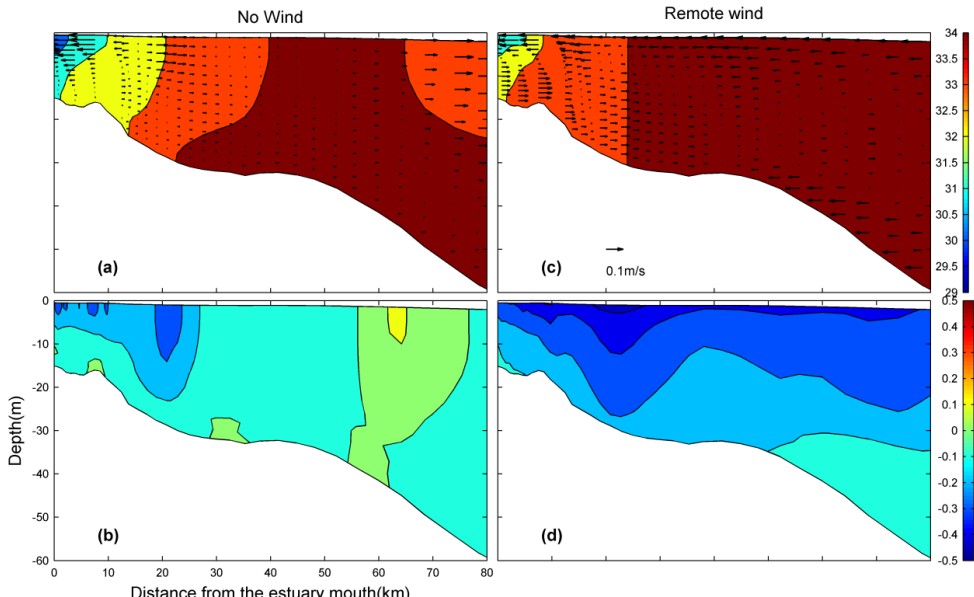

Figure 10. Comparison of 50-hour averaged current and salinity at the T4 section (on the continental shelf) during the spring tide: a) salinity and cross-shore current in Case 1; b) alongshore current (in m/s) in Case 1; c) salinity and cross-shore current in Case 3; d) alongshore current (in m/s) in Case 3.





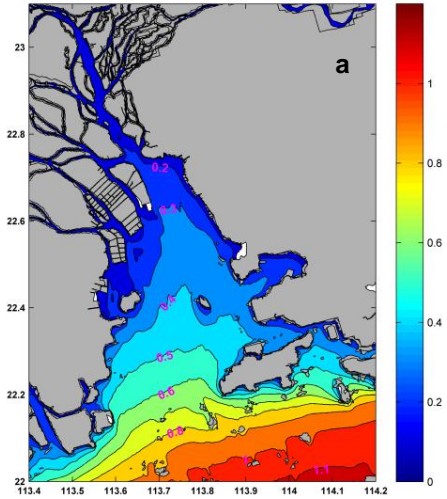

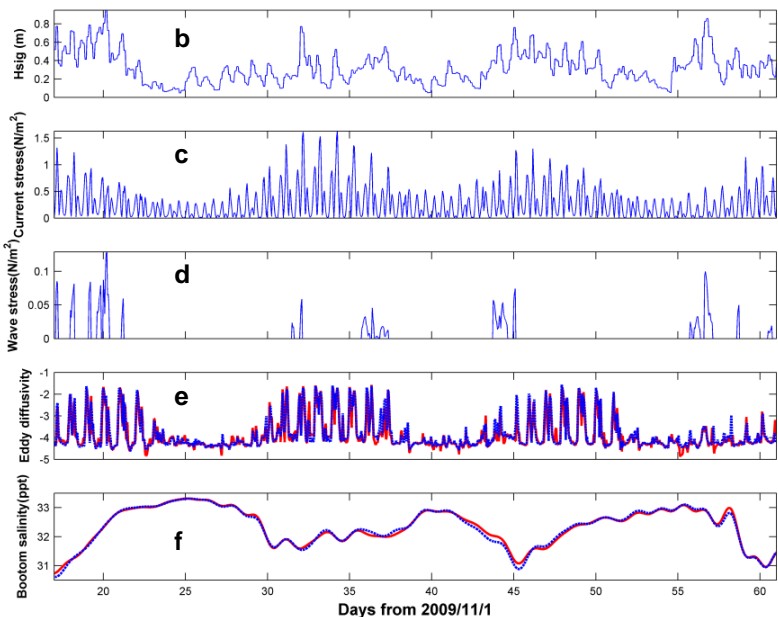

Figure 11. a) Spatial distribution of significant wave height (units: m) on Day 47; b) Time series of significant wave height at

5    the selected station; c) Current induced bottom stress; d) Wave-induced bottom stress; e) Vertically-averaged of the logarithm
(log10) of eddy diffusivity without (blue solid line) and with waves (red solid line). f) Bottom salinity without (blue solid
line) and with waves (red solid line).



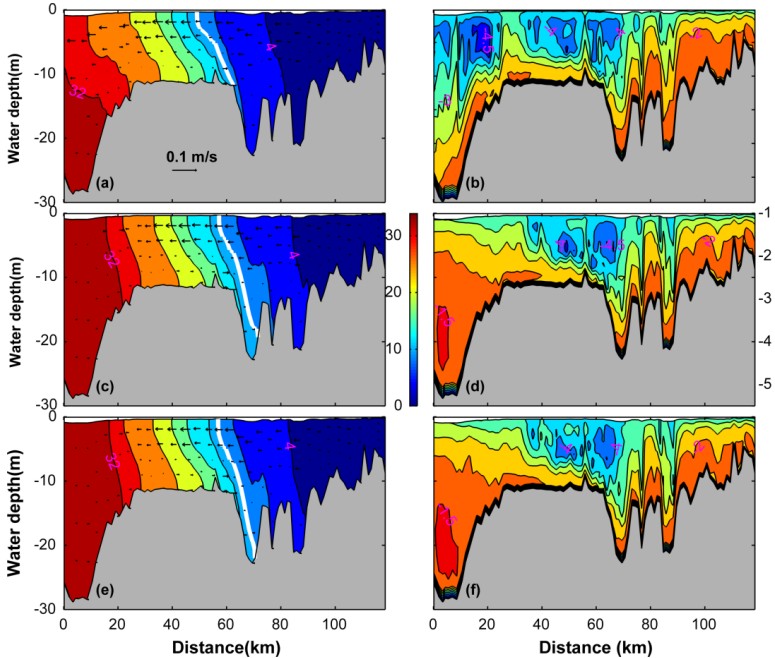

Figure 12. Comparisons of 50-hour averaged salinity, current and the log10 of eddy diffusivity along the longitudinal transect (T1) for Cases 1, 4 and 5. The three panels on the left, a), c) and e), show changes of salinity and current at T1 for Cases 1, 4 and 5, respectively. The three panels on the right, b), d) and f), show changes of the log10 of eddy diffusivity at T1 for Cases 1, 4 and 5, respectively.



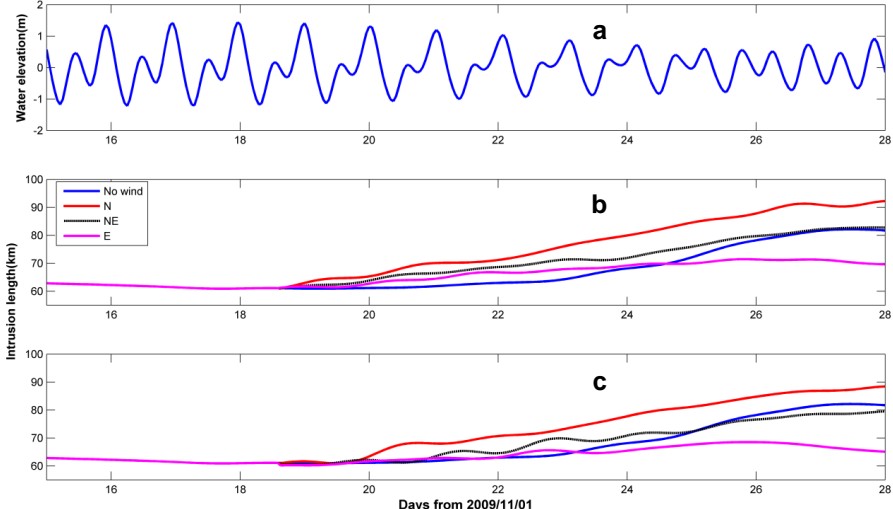

Figure 13. Comparison of lengths of the salt intrusion under different wind directions and the associated waves. a) Time series of water level; b) Low-pass filtered lengths of the salt intrusion under different wind directions; c) Low-pass filtered lengths of the salt intrusion under different wind directions and waves.





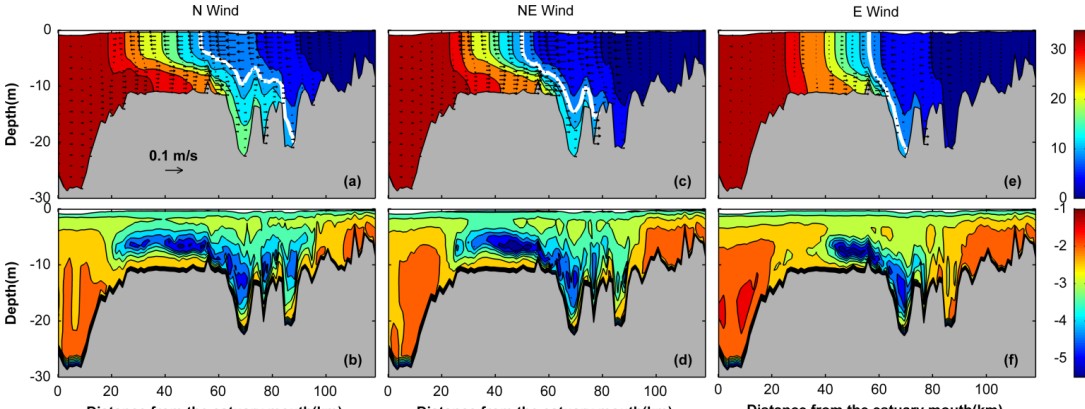

Figure 14. Comparison of 50-hour averaged salinity, current and eddy diffusivity along the longitudinal section (T1) under different wind directions. The white line shows the 10-ppt isohaline. The upper panel is for salinity and current, while the lower panel for the axial currents (in m/s).





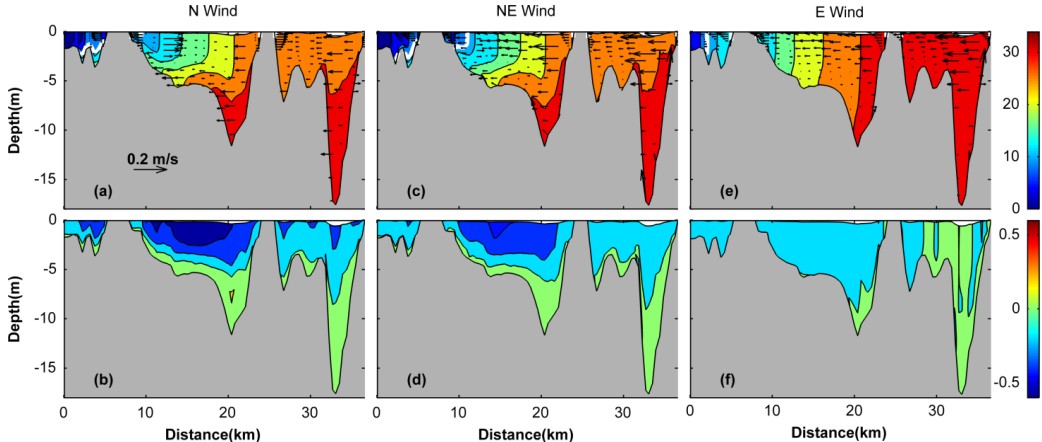

Figure 15. Comparison of 50-hour averaged salinity and current at the cross-section in the middle of the estuary (T4). The upper panel is for salinity and current, while the lower panel is for log10 of the eddy diffusivity.





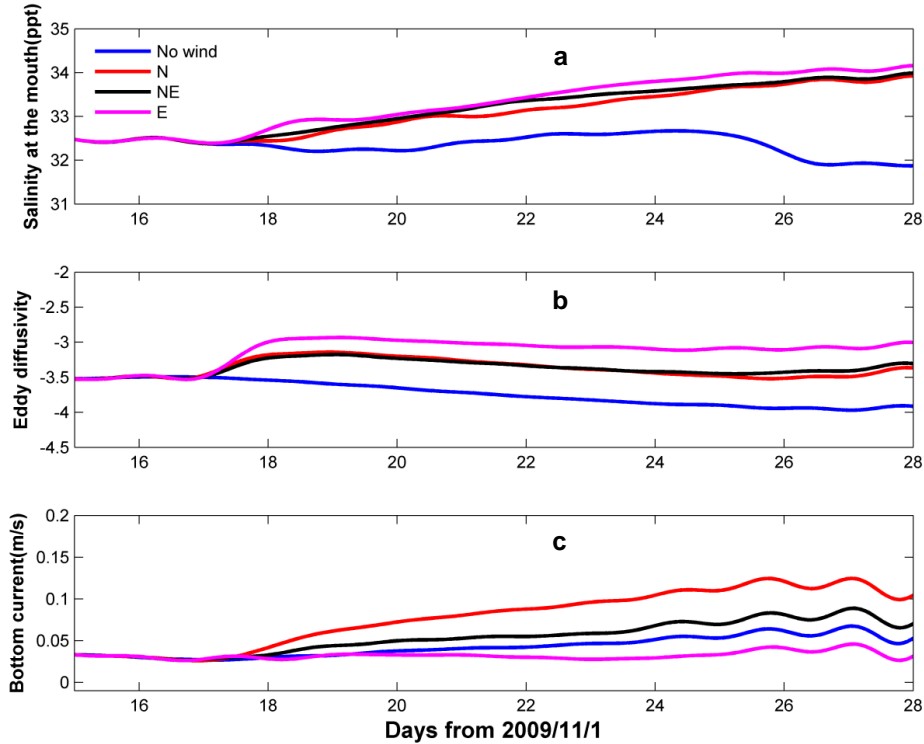

Figure 16. a) Low-pass filtered vertically-averaged salinity at the estuary mouth; b) Low-pass filtered log10 of mean eddy diffusivity along the longitudinal transect T1; c) Low-pass filtered mean bottom current along the T1.