# Peer review of "Effect of winds and waves on salt intrusion in the Pearl River Estuary"

_Ocean Science, 2017_

## Referee Comment (RC1) · Anonymous Referee #1 · 1 Oct 2017

This study investigated effects of winds and waves on salt intrusion in the Pearl River Estuary by conducting a series of numerical experiments. The topic is of interest and the results are reliable. Therefore, I recommend publication this manuscript in Marine Science after moderate revision.

The numerical model was validated using two observation data sets. Generally the model skills are low, particularly for the data set from Dec. 9 to 26, 2009. I understand that the bathymetry of Pearl River Estuary is rather complicated and to obtain a good performance of model simulation is not easy. However, the conclusion of "well performance" (see page 6, line 12 and line 29) is beyond the truth.

In the experiment without winds, the higher surface salinity during the spring was attributed to "lower freshwater inflow during the spring tide than during the neap tide

(page 7, line 27-28). However, the fact is that the river discharge of the Pearl River was higher during the spring than neap tides (see Fig. 3b).

Page 8, line 27 states "Generally, the local wind causes the landward salt flux to decrease", while line 29 states "overall, the total salt transport flux is seaward and decreased considerably under the local wind". Did local winds cause decrease in both landward and seaward salt flux? The two conclusions look contradictory.

In the experiment with remote wind, the subtidal water level increased across the entire domain (page 9 line 15). Does this mean the model failed to keep mass balance? Sea-level rise fills saltwater into the estuary and enhances salt intrusion. Does this filling effect affect the conclusion draw from the wind experiment?

The study found that the wind-induced water flux is larger during the spring (Day 46-48) than neap (Day 41-43) tides (page 9 line 25-26), and attributed this to tidal argument. I noticed that the winds were much stronger during the spring than neap tides (see Fig. 3 a). That accurately explains the change in wind-induced water flux from the spring to neap tides.

Figure 10, "T4" should be "T3".

Figure 15, Adding the contour of "0" might help identify landward and seaward flows.
* * *

---

## Referee Comment (RC2) · Anonymous Referee #2 · 13 Nov 2017

This is a very interesting and well written paper that examines the response of the Pearl River Estuary (velocity, salinity and salt intrusion) to freshwater inputs, winds and waves. The figures are of a high quality and the tables helpful to the overall study. The approach used was to apply well established models with actual environmental forcing variables to 'with-and-without' simulations. The COAWST system of models was used. The results demonstrate some very interesting (if sometimes intuitively obvious) phenomena that are worthy of publication. However, it is very difficult to quantitatively review an article of this nature, which is so far from 'first principles' and which relies so heavily on the interpretation of model results. The paper is also complicated in that numerous scenarios are explored and quite a lot is expected of the reader. A further complication is that actual environmental conditions are used. It is not easy to compare

e.g. the tidally-induced effects of neap and spring tides when runoff, wind and wave conditions are different for the chosen tides.

I have two important 'issues':

First, the authors refer to the 'skill' of the model (from Eq. 3 on p. 6) as being good or excellent - whereas a simple visual appraisal of Fig. 2 shows that the description (except for Fig. 2 (s,t,u)) should be 'satisfactory for the present exploratory purposes' Second, I am surprised that a major conclusion of the paper for the spring-neap behavior is apparently based on an error (p. 13, 35-40) – neap tides (days 40-42) are plotted to have much smaller freshwater runoff than spring tides (days 47-49) unless the plot (Fig. 3) is wrong or the intrusion depends only on the East River inflow.

I have four small issues

1. Eq. 1 and 2, p. 5 - the paper by Lerczak et al. (2006) does not use this decomposition as I recall – either use the correct citation or give more details of the derivation please 2. Eq. 3 on p. 6 uses an over-bar to denote e.g. an ensemble average, whereas the over-bar on velocity, U, on Line 8, refers to a depth-averaged quantity (also Lines 9 etc.) – please be consistent 3. Line 35 on p. 9 has 'T3' whereas the Fig. 10 caption has 'T4' (and is wrong) 4. Line 40 on p. 12 should have 'continuous increases'

---

## Referee Comment (RC3) · Anonymous Referee #3 · 15 Nov 2017

Although salinity intrusion under different hydrological and tide forcing conditions has been fully studied for many estuaries, wind and wave effects, especially wave effects, have not. This study focuses on the wind and wave effects on salinity intrusion in the Pearl River Estuary, which provides good information for the impacts of wind and waves on salinity intrusion in an estuary. The model has been calibrated using two observations. Overall, the model skills are acceptable for this study. The authors conducted a series of model diagnostic studies for the impact of local and remote winds with different directions as well as the change of transport processes associated with the change of wind, which provide good information for understanding the underlying processes.

Figure 7 compared different mechanisms for salt intrusion, which shows the different processes associated with different wind forcings, i.e., remote and local winds, in

[Figure]

particular, the tidal oscillatory transport and steady shear are different with respect to remote and local winds. However, they were not clearly discussed. It will be good if the authors can incorporate these into the discussion sections.

As there are many model scenarios with different conditions, the paper is not easy to follow and some materials can be removed (i.e., the discussion of salinity at the mouth as the change of salinity at the mouth can be seen from other figures).

Some figure captions need some revision to make them more clear. i.e., Fig. 14. Are currents and axial currents are same?

---

## Author Comment (AC2) · 29 Nov 2017

Comment (1) from referee #2 This is a very interesting and well written paper that examines the response of the Pearl River Estuary (velocity, salinity and salt intrusion) to freshwater inputs, winds and waves. The figures are of a high quality and the tables helpful to the overall study. The approach used was to apply well established models with actual environmental forcing variables to 'with-and-without' simulations. The COAWST system of models was used. The results demonstrate some very interesting (if sometimes intuitively obvious) phenomena that are worthy of publication. However, it is very difficult to quantitatively review an article of this nature, which is so far from 'first principles' and which relies so heavily on the interpretation of model results. The paper is also complicated in that numerous scenarios are explored and quite a lot is

expected of the reader. A further complication is that actual environmental conditions are used. It is not easy to compare e.g. the tidally-induced effects of neap and spring tides when runoff, wind and wave conditions are different for the chosen tides.

Response:

We very appreciate the reviewer's efforts for providing us valuable comments.

Comment (2) from referee #2

I have two important 'issues': First, the authors refer to the 'skill' of the model (from Eq. 3 on p. 6) as being good or excellent - whereas a simple visual appraisal of Fig. 2 shows that the description (except for Fig. 2 (s,t,u)) should be 'satisfactory for the present exploratory purposes'

Response:

We modify this part of the text, not to exaggerate the model's performance. As also raised by the first reviewer, the worse model performance occurs in the dataset of Dec.9 to 26, 2009, which was conducted in the Modaomen Estuary. This estuary is narrower, and features complicated geometry and bathymetry. The model's resolution is not high enough to resolve all the small scale variations, resulting in a poor performance.

Comment (3) from referee #2

Second, I am surprised that a major conclusion of the paper for the spring-neap behavior is apparently based on an error (p. 13, 35-40) – neap tides (days 40-42) are plotted to have much smaller freshwater runoff than spring tides (days 47-49) unless the plot (Fig. 3) is wrong or the intrusion depends only on the East River inflow.

Response: This is a confusion caused by our insufficient explanation. The river discharge data shown in Fig. 3b are those from the upstream of the West, North and East Rivers. The freshwater takes approximately 3-5 days to reach the head of the PRE. Thus the river inflow into the PRE lags the variations of upstream river discharge.

Therefore, the neap tide coincides to a higher inflow, while the spring tide to a lower inflow. Another fact is that under the similar river discharge, more freshwater is detained in the Pearl River Network during the spring tide by strengthened bottom friction, thereby smaller amount of river inflow into the estuary is expected, and vice versa. Above explanation are added into the revised manuscript in the end of the third paragraph of section 5.1. To further confirm our statement, we select the cross-section at the HumenOutlet, and calculate the freshwater flux during our study period. More information can be seen in the supplement pdf file.

Comment (4) from referee #2

I have four small issues 1. Eq. 1 and 2, p. 5 - the paper by Lerczak et al. (2006) does not use this decomposition as I recall – either use the correct citation or give more details of the derivation please.

Response: Our derivation should be right. Lerczak et al. (2006)'s equation: Fs=<âĽňâŰŠãĂŰus dAãĂŮ> , the equation has only one argument, but has two integral symbols. This should be not right. We only correct this. as for the equation 2, we are similar to that used by Chen and Sanford (2009): Axial Wind Effects on Stratification and Longitudinal Salt Transport in an Idealized, Partially Mixed Estuary. Journal of Physical Oceanography 39 (8) :1905-1920.

Comment (5) from referee #2

2. Eq. 3 on p. 6 uses an over-bar to denote e.g. an ensemble average, whereas the over-bar on velocity, U, on Line 8, refers to a depth-averaged quantity (also Lines 9 etc.) – please be consistent

Response: We use âŇľOâŇł for the timely mean, and the overbar for the depth-averaged. We revise the text correspondingly.

Comment (6) from referee #2

3. Line 35 on p. 9 has 'T3' whereas the Fig. 10 caption has 'T4' (and is wrong)

Response: We correct this error in the caption of Fig. 10.

Comment (7) from referee #2

4. Line 40 on p. 12 should have 'continuous increases'

Response: We correct this error in the revised manuscript.

Please also note the supplement to this comment:
https://www.ocean-sci-discuss.net/os-2017-73/os-2017-73-AC2-supplement.pdf

[Figure]

**Supplement:**

Supplement to the comment (3) from referee #2:

To further confirm our statement, we select the cross-section at the Humen Outlet, and calculate the freshwater flux during our study period. The freshwater flux is calculated as:

$$\text{Flux} = \int \left(1 - \frac{s}{s_0}\right) * u \, dA$$

where $s_o$ is set as 32 ppt. We obtained the instantaneous and subtidal (by a low-pass filter with cutoff period of 34 hours). The location of the Humen section and the freshwater flux are shown in the following figures:

[Figure]

[Figure]

Here the positive means landward flux (in m$^3$/s). From the results, we can clearly see that the subtidal freshwater fluxes during neap tides are larger than those during spring tides, e.g. the neap tide of Day 42-43 is larger than the spring tide of Day 46-48.

---

## Author Response (AR1)

This study investigated effects of winds and waves on salt intrusion in the Pearl River Estuary by conducting a series of numerical experiments. The topic is of interest and the results are reliable. Therefore, I recommend publication this manuscript in Marine Science after moderate revision.

Response:

We are grateful for the reviewer's encouragement and providing us constructive comments.

The numerical model was validated using two observation data sets. Generally the model skills are low, particularly for the data set from Dec. 9 to 26, 2009. I understand that the bathymetry of Pearl River Estuary is rather complicated and to obtain a good performance of model simulation is not easy. However, the conclusion of "well performance" (see page 6, line 12 and line 29) is beyond the truth.

Response:

We totally agree with the reviewer's comment. The dataset from Dec. 9 to 26, 2009 were obtained in the Modaomen Estuary, another estuary in the Pearl River Delta, which has very complex geometry and bathymetry, and a much small size. The model's spatial resolution is not fine enough to resolve the small scale variations of geometry and bathymetry, thus the model performance is not good enough. We revise this statement to be "performs satisfactorily", "agree reasonably" and "provides acceptable simulations", respectively (P6 L12, L30 and P7 L1 in the revised manuscript without track changes).

In the experiment without winds, the higher surface salinity during the spring was

attributed to "lower freshwater inflow during the spring tide than during the neap tide (page 7, line 27-28). However, the fact is that the river discharge of the Pearl River was higher during the spring than neap tides (see Fig. 3b).

Response:

This is a confusion caused by our insufficient explanation. The river discharge data shown in Fig. 3b are those from the upstream of the West, North and East Rivers. These freshwater takes approximately 3-5 days to reach the head of the PRE. Thus the river inflow into the PRE lags the variations of upstream river discharge. Therefore, the neap tide coincides to a higher inflow, while the spring tide to a lower inflow. Another fact is that under the similar river discharge, more freshwater is detained in the Pearl River Network during the spring tide by strengthened bottom friction, thereby smaller amount of river inflow into the estuary is expected, and vice versa. Above explanation are added into the revised manuscript in the end of the third paragraph of section 5.1 (P7 L29-32 in the revised manuscript without track changes). To further confirm our statement, we select the cross-section at the HumenOutlet, and calculate the freshwater flux during our study period. The freshwater flux is calculated as:

$$\text{Flux} = \int \left(1 - \frac{s}{s_0}\right) * u \, dA$$

where $s_o$ is set as 32 ppt. We obtained the instantaneous and subtidal (by a low-pass filter with cutoff period of 34 hours). The location of the Humen section and the freshwater flux are shown in the following figures:

[Figure]

Here the positive means landward flux (in m$^3$/s). From the results, we can clearly see that the subtidal freshwater fluxes during neap tides are larger than those during spring tides, e.g. the neap tide of Day 42-43 is larger than the spring tide of Day 46-48.

Page 8, line 27 states "Generally, the local wind causes the landward salt flux to

decrease", while line 29 states "overall, the total salt transport flux is seaward and decreased considerably under the local wind". Did local winds cause decrease in both landward and seaward salt flux? The two conclusions look contradictory.

Response:

This is a misunderstanding. In the manuscript, we state that "Generally, the local wind causes the landward tidal flux to decrease", not the "salt flux to decrease". The tidal flux is "the tidal oscillatory flux", one component of "the salt flux". Our statement is to say that the tidal oscillatory flux is decreased while the total salt flux is increased by the local wind.
We correct this by replacing "the tidal flux" with "the tidal oscillatory flux" in the revised manuscript (P8 L32 in the revised manuscript without track changes).

In the experiment with remote wind, the subtidal water level increased across the entire domain (page 9 line 15). Does this mean the model failed to keep mass balance?

Response:

This does not mean the model failed to keep mass balance. As we impose a water level setup at the open boundary, more saline water is pumped into the estuary. In the meantime, freshwater is discharged into the estuary. These water inflows elevate the water level in the estuary, and keep the mass in balance. The water level setup occurs in our study period. However, when the water level at the open boundary recedes, the accumulated water in the estuary will flow out.

Sea level rise fills saltwater into the estuary and enhances salt intrusion. Does this filling effect affect the conclusion drawn from the wind experiment?

Response:

Sea level rise fills saltwater into the estuary and enhances salt intrusion. It definitely sets a different background for the wind effects. However, the conclusions drawn from the wind experiments still hold true. The local NE wind drives more landward salt transport flux and pushes the river plume against the western shore, the remote wind mixes the river plume water near the estuary mouth and pump saltier water into the estuary by Ekman transport. These conclusions will not change under sea level rise, though the magnitudes may change more or less. As seen in our simulation scenarios, Case 4 is a combination of cases 2 and 3, and the salt intrusion in case 4 is simply a summation of those in case 2 and 3. Note that in case 3, we include a water level setup, similar to sea level rise. We can postulate that the integrated results of winds and sea level rise can be a summation of their effects.

The study found that the wind-induced water flux is larger during the spring (Day 46-48) than neap (Day 41-43) tides (page 9 line 25-26), and attributed this to tidal argument. I noticed that the winds were much stronger during the spring than neap tides (see Fig.3a). That accurately explains the change in wind-induced water flux from the spring to neap tides.

Response:
Of course, the wind is stronger during the spring tide than during the neap tide, and drives more saltier water into the estuary in the spring tide. In the meantime, the spring tide induces more Stokes inflow flux, and thus augment the wind-induced flux. So the tidal effect is an extra, but may not dominate. We did not distinguish these two effects.

Figure 10, "T4" should be "T3".

Response:

We correct the error by replacing "T4" with "T3" in the caption of Fig. 10 (P30 L2 in the revised manuscript without track changes).

Figure 15, Adding the contour of "0" might help identify landward and seaward flows.

Response:

We redraw this Fig. 15 with the contour of "0" and revised the caption accordingly (P35 in the revised manuscript without track changes).

**Anonymous Referee #2**

This is a very interesting and well written paper that examines the response of the Pearl River Estuary (velocity, salinity and salt intrusion) to freshwater inputs, winds and waves. The figures are of a high quality and the tables helpful to the overall study. The approach used was to apply well established models with actual environmental forcing variables to 'with-and-without' simulations. The COAWST system of models was used. The results demonstrate some very interesting (if sometimes intuitively obvious) phenomena that are worthy of publication. However, it is very difficult to quantitatively review an article of this nature, which is so far from 'first principles' and which relies so heavily on the interpretation of model results. The paper is also complicated in that numerous scenarios are explored and quite a lot is expected of the reader. A further complication is that actual environmental conditions are used. It is not easy to compare e.g. the tidally-induced effects of neap and spring tides when runoff, wind and wave conditions are different for the chosen tides.

Response:

We very appreciate the reviewer's efforts for providing us valuable comments.

I have two important 'issues':
First, the authors refer to the 'skill' of the model (from Eq. 3 on p. 6) as being good or excellent - whereas a simple visual appraisal of Fig. 2 shows that the description (except for Fig. 2 (s,t,u)) should be 'satisfactory for the present exploratory purposes'

Response:

We modify this part of the text, not to exaggerate the model's performance (P6 L12, L30 and P7 L1 in the revised manuscript without track changes). As also raised by the first reviewer, the worse model performance occurs in the dataset of Dec.9 to 26, 2009, which was conducted in the Modaomen Estuary. This estuary is narrower, and features complicated geometry and bathymetry. The model's resolution is not high enough to resolve all the small scale variations, resulting in a poor performance.

Second, I am surprised that a major conclusion of the paper for the spring-neap behavior is apparently based on an error (p. 13, 35-40) – neap tides (days 40-42) are plotted to have much smaller freshwater runoff than spring tides (days 47-49) unless the plot (Fig. 3) is wrong or the intrusion depends only on the East River inflow.

Response:
This is a confusion caused by our insufficient explanation. The river discharge data shown in Fig. 3b are those from the upstream of the West, North and East Rivers. The freshwater takes approximately 3-5 days to reach the head of the PRE. Thus the river inflow into the PRE lags the variations of upstream river discharge. Therefore, the neap tide coincides to a higher inflow, while the spring tide to a lower inflow. Another fact is that under the similar river discharge, more freshwater is detained in the Pearl River Network during the spring tide by strengthened bottom friction, thereby smaller amount of river inflow into the estuary is expected, and vice versa.

Above explanation are added into the revised manuscript in the end of the third paragraph of section 5.1 (P7 L29-32 in the revised manuscript without track changes). To further confirm our statement, we select the cross-section at the Humen Outlet, and calculate the freshwater flux during our study period. The freshwater flux is calculated as:

$$\text{Flux} = \int \left(1 - \frac{s}{s_0}\right) * u \, dA$$

where $s_o$ is set as 32 ppt. We obtained the instantaneous and subtidal (by a low-pass filter with cutoff period of 34 hours). The location of the Humen section and the freshwater flux are shown in the following figures:

[Figure]

[Figure]

Here the positive means landward flux (in m$^3$/s). From the results, we can clearly see that the subtidal freshwater fluxes during neap tides are larger than those during spring tides, e.g. the neap tide of Day 42-43 is larger than the spring tide of Day 46-48.

I have four small issues
1. Eq. 1 and 2, p. 5 - the paper by Lerczak et al. (2006) does not use this decomposition as I recall – either use the correct citation or give more details of the derivation please.

Response:

Our derivation should be right. Lerczak et al. (2006)'s equation:

Fs $=< \iint$ us dA $>$,the equation has only one argument, but has two integral symbols. This should be not right. We only correct this. as for the equation 2, we are similar to that used by Chen and Sanford (2009): Axial Wind Effects on Stratification and Longitudinal Salt Transport in an Idealized, Partially Mixed Estuary. Journal of

Physical Oceanography 39 (8) :1905-1920.

2. Eq. 3 on p. 6 uses an over-bar to denote e.g. an ensemble average, whereas the over-bar on velocity, U, on Line 8, refers to a depth-averaged quantity (also Lines 9 etc.) – please be consistent

Response:

We use $\langle 0 \rangle$ for the timely mean, and the overbar for the depth-averaged. We revise the text correspondingly (P6 L4-5 in the revised manuscript without track changes).

3. Line 35 on p. 9 has 'T3' whereas the Fig. 10 caption has 'T4' (and is wrong)

Response:

We correct this error in the caption of Fig. 10 (P30 L2 in the revised manuscript without track changes).

4. Line 40 on p. 12 should have 'continuous increases'

Response:

We correct this error in the revised manuscript (P13 L2 in the revised manuscript without track changes).

**Anonymous Referee #3**

Although salinity intrusion under different hydrological and tide forcing conditions has been fully studied for many estuaries, wind and wave effects, especially wave effects, have not. This study focuses on the wind and wave effects on salinity intrusion in the Pearl River Estuary, which provides good information for the impacts

of wind and waves on salinity intrusion in an estuary. The model has been calibrated using two observations. Overall, the model skills are acceptable for this study. The authors conducted a series of model diagnostic studies for the impact of local and remote winds with different directions as well as the change of transport processes associated with the change of wind, which provide good information for understanding the underlying processes.

Response:

We thank for the reviewer's constructive comments.

Figure 7 compared different mechanisms for salt intrusion, which shows the different processes associated with different wind forcings, i.e., remote and local winds, in particular, the tidal oscillatory transport and steady shear are different with respect to remote and local winds. However, they were not clearly discussed. It will be good if the authors can incorporate these into the discussion sections.

Response:

We have already explained the changes in the steady shear transport flux by the local winds (page 8 line 38 to page 9 line 6) and by the remote winds (page 9 line 30-31). The local NE winds change the estuarine circulation from vertically sheared to horizontally segregated, and change the salinity distribution from vertically stratified to horizontally differentiated, therefore increase the steady shear transport flux. Under other wind directions, such as the SE wind, the wind mixing may decrease the estuarine circulation and salinity stratification, and thus the steady shear transport. The remote wind is to increase the mixing at the estuary's mouth by downwelling current, with an effect similar to tidal straining during flood tides, and to decrease the estuarine circulation, thus generally decrease the steady shear transport flux.

For the tidal oscillatory part, the remote wind is to decrease the magnitude of the tidal fluctuations of salinity due to more mixing, while for the local wind, the mechanisms

seem difficult to analyze. Overall, the tidal oscillatory flux is an insignificant component in the total salt transport flux, we do not further explore it.

As there are many model scenarios with different conditions, the paper is not easy to follow and some materials can be removed (i.e., the discussion of salinity at the mouth as the change of salinity at the mouth can be seen from other figures).

Response:

We appreciate reviewer's comment. After careful consideration, we choose to keep the change in salinity at the mouth in the discussion. Though other figures show salinities at the mouth, but it is difficult to compare them under different scenarios. Moreover, Figure 16a show the temporal evolution of the salinity at the mouth, and links closely to the time series of diffusivity and bottom current along the longitudinal transect. These processes need to be combined together for explaining the effects of different winds.

Some figure captions need some revision to make them more clear. i.e., Fig. 14. Are currents and axial currents are same?

Response:

This is a typo. We make a distinction of lateral and axial currents at the top and bottom panels, respectively (P34 in the revised manuscript without track changes).

**List of all relevant changes made in the manuscript**

(1) P6 L4-5 in the revised manuscript without track changes. Eq. (3) are modified. $\overline{O}$ is changed to $\langle O \rangle$.

(2) P6 L12 and L30. "Performs well" and "agree reasonably well" are changed to "performs satisfactorily" and "agree reasonably", respectively.

(3) P7 L1. "Provides good simulation" is changed to "provides acceptable simulation".

(4) P7 L29-32. The following text are added: It should also be noted that the upstream river discharge in the West, North and East Rivers takes 3-5 days to reach the estuary's head, thus the river inflow into the PRE lags the variations of the upstream river discharge. Therefore, in our study period, the neap tide (Day 41-43) corresponds to a higher inflow, while the spring tide (Day 46-48) coincides with a lower inflow.

(5) P8 L32. "Tidal flux" is changed to "tidal oscillatory flux".

(6) P13 L2. "Continuously" is changed to "continuous".

(7) P14 L29. Acknowledgements to three anonymous reviewers are added.

(8) P30 L2. "T4" in the figure caption of Fig. 10 is changed to "T3".

(9) P34 L3-4. The figure caption of Fig. 14 is modified according to the reviewers' comments.

(10) P35. Figure 15 and its' caption are modified according to the reviewers' comments.

[revised manuscript text omitted]